



# Migrating and Non-Migrating Tides Observed in the Stratosphere from FORMOSAT-3/COSMIC Temperature Retrievals

Uma Das[1*], William Ward[2], Chen Jeih Pan[3], Sanat Kumar Das[4]

[1]Indian Institute of Information Technology Kalyani, Kalyani, West Bengal, India
[2]Department of Physics, University of New Brunswick, Fredericton, Canada
[3]Institute of Space Science, National Central University, Zhongli, Taiwan
[4]Environmental Sciences Section, Bose Institute, Kolkata, India

*Correspondence to*: Uma Das (uma@iiitkalyani.ac.in)

**Abstract.** FORMOSAT-3/COSMIC temperature data during 2009 to 2010 are analysed for tides in the middle atmosphere
from ~10 to 50 km. COSMIC is a set of six micro satellites in near sun synchronous orbits with 30° orbital separations and
provides good phase space sampling of tides. Short term tidal variability is deduced by considering ±10 days' data together.
The DW1 tide is found to peak over the equator at 30 km. It maximises and slightly shifts poleward during winters and thus
is attributed to ozone absorption. Over mid and high latitudes, DW1 and the non-migrating tides DS0 and DW2 are
intermittent in nature. Numerical experiments in the current study show that these could be a result of aliasing as they are
found to occur at times of steep rise or fall in the mean temperature, particularly during the SSW of 2010. Further, stationary
planetary wave components are found to be of very large amplitudes in the northern hemispheres reaching 18K at 30 km
over 65°N. By using data from COSMIC over shorter durations, aliasing between SPW and non-migrating tides is reduced
and thus results in the large amplitudes of the former. This study clearly indicates that non-linear interactions are not a very
important source of generation of the non migrating tides in the high latitude winter hemisphere. There is also a modulation
of SPW1 by ~60 days in the high latitudes, which was not seen earlier.

## 1 Introduction

Tidal variability in the atmosphere is a very important parameter to understand the long term as well as day to day variations
in the atmosphere. Till date the short term global tidal variabilities in the middle atmosphere have not been identified due to
lack of sufficient data. Using only ground based data the dominant tidal periods can be identified but it is difficult to obtain
the longitudinal variability (i.e., wavenumber of the tides) unless there are simultaneous measurements at different
longitudes along the same latitude circle [For e.g., *Q Wu et al.*, 2008]. Even if such measurements are possible over a given
latitude, all latitudes of the globe cannot be covered due to various reasons including land-sea differences. On the other hand,
satellites have the ability to take global measurements; however, their local time coverage is different. For example, TIMED
satellite, which is in a near-Sun synchronous orbit, takes ~60 days to cover all local times at a given location [*Mertens et al.*,
2004; *Remsberg et al.*, 2003; *Remsberg et al.*, 2008]. This implies that to derive tidal characteristics, data has to be





accumulated for ~60 days [*Remsberg et al.*, 2008; *Sakazaki et al.*, 2012; *Xu et al.*, 2014; *Zhang et al.*, 2006]. Even then, due to the satellite's orbit, noon time observations are not available. Thus all phases of the tides, specifically migrating tides, are not sampled. This poses a problem for accurate determination of tidal variabilities. Accumulating data over 60 days also means that the short term variabilities are lost. Further, any changes in the mean variation of the temperature aliases into the

energy of migrating tides [*Forbes et al.*, 1997; *Sakazaki et al.*, 2012].

Tides are produced due to absorption of solar radiation by water vapour in the troposphere and ozone in the stratosphere and also due to latent heat release in the troposphere. There are also tides produced in situ in the thermosphere due to EUV absorption. The tides that move westward with apparent motion of the Sun are called the migrating tides. The migrating diurnal tidal characteristics in the stratosphere have been retrieved using temperature retrievals from FORMOSAT-

3/COSMIC mission [*Anthes et al.*, 2008] using monthly data for the period 2007-2008 [*Pirscher et al.*, 2010]. Maximum amplitudes of 0.8 to 1.0 K were found over the tropics at 30 km altitude and are attributed to ozone absorption. There are several papers in literature that describe the theory [*Chapman and Lindzen*, 1970; *Forbes and Garrett*, 1979] and observed characteristics of tides at various altitudes in the stratosphere, mesosphere and thermosphere from ground based measurements of radars, lidars, etc., [*Liu et al.*, 2007; *Pancheva and Mukhtarov*, 2000; *She et al.*, 2004; *Xue et al.*, 2007],

satellite observations of TIDI and SABER instruments onboard TIMED [*Mukhtarov et al.*, 2009; *Qian Wu et al.*, 2006], from UARS [*Shepherd et al.*, 2012; *D L Wu et al.*, 1998], MLS [*D L Wu and Jiang*, 2005], reanalysis [*Gan et al., 2014*] and model datasets [*Sakazaki et al.*, 2018]. Based on results obtained from TIMED tidal diagnostics, the Climatological Tidal Model of the Thermosphere (CTMT) constituting the most important diurnal and semidiurnal tides have been proposed [*Oberheide et al.*, 2011a]. Using global cloud imagery, the Global Scale Wave Model was developed for tides arising due to

latent heat releases [*Hagan*, 2002; 2003]. Such models are further used as parameterizations for other global circulation models in the lower and upper atmosphere.

There are also the non migrating tides in the atmosphere whose apparent motion is either slow or faster than the Sun. Some of these tides are thought to be produced due to non linear interactions between stationary planetary waves (SPW) and migrating tides. However, significant debate exists if the non migrating tides are truly a geophysical phenomenon or are an

artefact of the method of analysis. It was proposed that SPW of wavenumber 1 (SPW1) interacts non-linearly with diurnal migrating tide (DW1) and results in the non migrating tides DS0 and DW2 [The notation of the tides is as follows: First letter indicates the period of the tide - D for diurnal, S for semi-diurnal, T for terdiurnal; second letter indicates if the tide is westward (W) or eastward (E) propagating or stationary (S), and finally the last character is a digit which gives the wavenumber of the tide. The same notation will be followed for the rest of the paper]. Similarly, SPW of wavenumber 1

(SPW1) interacts with semi-diurnal migrating tide (SW2) and produces SW1 and SW3. Many studies support this school of thought based on correlation studies [*Xu et al.*, 2014]. However, it is also a possibility that the high correlation is observed because of aliasing between these different components.

Among the non migrating tides, a reasonably well understood tide is DE3 (eastward propagating diurnal tide of wavenumber 3). The observation of the wave-4 structure in the equatorial ionisation anomaly of the ionosphere due to DE3 tide is one of





the most important discoveries of the last decade [*Immel et al.*, 2006]. The DE3 tide is very unique to the Earth and is produced in the troposphere due to the specific distribution of the landmasses and oceans and associated heating [*Oberheide et al.*, 2011b]. As the tide propagates upwards it modifies the various atmospheric parameters and this emphasizes the importance of troposphere-ionosphere coupling and also the need for obtaining the short term tidal variabilities.

The various tides generated in the lower atmosphere propagate upward, grow in amplitude and affect the large scale

dynamics, chemistry and energetics of the thermosphere and ionosphere and thus accurate determination of the variability of these various tides and other waves at the point of generation is extremely important to understand the atmosphere. In the current study temperature data from FORMOSAT-3/COSMIC during 2009 to 2010 is analysed to extract migrating and non migrating tides and stationary planetary waves globally over shorter time periods of ±10 days. Along with obtaining short term variability in the said tides, the paper also addresses the aliasing involved between (1) mean temperature and migrating

tides and (2) stationary planetary waves and non migrating tides, particularly in the high latitudes.

## 2 Data

COSMIC is a constellation of six micro satellites working on the principle of GPS Radio Occultation (GPS RO). It involves active Earth limb sounding by radio transmissions by GPS satellites at 20,200 km and are observed by the COSMIC satellites in low Earth orbits [*Anthes et al.*, 2008]. The phase delay in signal received due to change in refractivity is

converted to electron density in the ionosphere and temperature and other parameters in the lower atmosphere [*Kuo et al.*, 2004]. This technique provides a near-vertical scan of the atmosphere with good vertical resolution, global coverage, insensitivity to atmospheric particulate matter [*Kuo et al.*, 2004; *Kursinski et al.*, 1997]. The six satellites are placed in ~800 km orbits with 30° separations. This enables the local time coverage of all satellites, taken together, theoretically over any given location to be possible in approximately 10 days. In this way, COSMIC satellites have a huge advantage over SABER

in terms of global coverage. However, the altitude coverage of COSMIC is from surface to 60 km (atmPrf, dry temperature data product) with temperatures data reliable up to 50 km over the equator and further lower over mid and high latitudes [*Das and Pan*, 2014], while SABER has coverage from 20 to 120 km, and thereby the latter enables studies of stratosphere, mesosphere and lower thermosphere. Thus the data from COSMIC can only be used for tropospheric and stratospheric studies (and the ionospheric data products can be used for ionospheric studies). In the current study, dry temperature

'atmPrf' profiles from the lower atmospheric data from FORMOSAT-3/COSMIC are analysed. COSMIC temperatures are smaller by 2 to 3 K than SABER temperatures across all latitudes below 0.3 hPa and larger above this altitude. The agreement of COSMIC temperatures with Aura-MLS observations is much better and in the range of ±1 K upto 2 hPa [*Das and Pan*, 2014].



## 2.1 Satellite Sampling and Phase Space:

The data obtained from COSMIC using the technique of GPS RO are not regular, i.e., the retrieved data are not uniformly spaced in space and time. This non-uniform and pattern-less sampling is in a way supportive to the current study to characterise the variability of tides in the middle atmosphere. SABER onboard TIMED satellite is in a near Sun synchronous orbit and its observations are regular and in a clustered mode. Over a given location, the satellite observes the atmosphere at gradually reducing local times, and takes sixty days to cover the entire 24 hour diurnal duration. Due to this, to obtain

sampling at all local times using SABER, analysis has to be done over data accumulated for 60 days. Thus, the tidal parameters obtained are a temporal average representation over the sixty days considered. On the other hand, accumulating data for 60 days induces serious aliasing problems as the variation in average or mean temperature aliases into the diurnal tides' amplitudes and the energy of SPWs alias into that of the non migrating tides. These problems are addressed here using COSMIC data.

As mentioned earlier 10 days data from all six COSMIC satellites is required to cover the 24 local hour diurnal duration over any given location and thus short term tidal variability can be obtained. If data from only one satellite is considered one would require sixty days of data for tidal analysis, similar to SABER. To establish this aspect and to ascertain the necessary and sufficient conditions for the amount of data required for accurate tidal characteristic extraction, COSMIC data is considered as follows for the analysis. Data is divided in to two overlapping groups, consisting of four satellites each. First

group, named group 'G1', takes data from satellites C001, C002, C003, and C004 and the second group, named group 'G2', takes data from C004, C005, C006 and C001. In principle, we could have divided the satellites into groups of three satellites, however, due to technical problems, sometimes data from one or other satellite is not available entirely, or less data is available. To overcome this, we made groups of four, with two satellites in common. A third group consisting of all six satellites is also investigated; this is named group 'G0'. Further, data of each group is analysed by considering ±10 days' data

centred over each day. Differences observed in results obtained from G0, G1 and G2 would throw light upon the possible aliasing effects involved in arriving at erroneous results if data considered is more or less. Data from the C004 satellite is also analysed separately using the same method by considering data over ± 30 days.

When satellite data are considered for tidal analysis, to be able to analyse the data with minimal aliasing related problems, it is expected that the two dimensional space of universal time and longitude (over a certain latitude) is uniformly sampled by

the satellite. The same can be verified from a different perspective of total phase. The universal time ($t$) and longitude ($\lambda$) of each observation give the total phase ($2\pi f t + 2\pi s \lambda$) of each wave of frequency ($f$) and wave number ($s$) ranging from 0 to $2\pi$. If all phases of a given wave are sampled, i.e., if phase sampling is sufficiently uniform, then the characteristics of the wave, viz., amplitude, and phase, can be extracted with little uncertainty. To understand this, the total phase of the important wave component DW1, are investigated over equator and 65°N and shown in Figure 1 for the different groups G0, G1 and

G2 (by considering ±10 days' data) as well as for C004 satellite (by considering ±30 days' data). It can be seen that for both latitudes shown, the phase sampling is uniform on any given day for all the waves. The number of data points reduced in





general over the period investigated, due to reduction in overall number of observations. It can also be seen that the sampling is also uniform when data from one satellite (C004) was considered over 61 days. This phase space sampling is verified for all other waves of interest to the current study, and there is no change in the inference.

## 3 Analysis

Data in each group (G0/G1/G2/C004) are investigated using the least squares fitting technique. The following function is fit to the two dimensional temperature data at each altitude in universal time and longitude space, to include (a) mean, (b) diurnal, semi-diurnal and ter-diurnal tides with wave numbers from -4 to 4, where negative wave numbers denote eastward propagating tides and positive wave numbers denote westward propagating tides and (c) SPWs with wave numbers from 1 to 3.

$$T(t,\lambda) = T_0 + \sum_{i=1}^{3} \sum_{j=-4}^{4} T_{ij}\cos(2\pi f_i t + 2\pi s_j \lambda - \phi_{ij}) + \sum_{k=1}^{3} T_k \cos(2\pi s_k \lambda - \phi_k)$$

It may be noted that data from G0, G1, G2 are analysed using ±10 days' data and data from C004 is analysed using ±30 days' data. This equation results in 61 fitted parameters that are carefully investigated in the ensuing sections.

## 4 Tidal Characteristics

The mean temperature and amplitudes of DW1 and SPW1 at 30 km obtained from the analysis of temperature data during November 2009 – September 2010 are shown in the three rows of Figure 2, respectively. Each column indicates the results obtained from the three groups G0, G1, and G2 using ±10 days data and from satellite C004 using ±30 days data. The last column shows the numerical difference between results obtained from group G0 and C004. It can be seen that the results obtained from the three groups are very similar, with minute differences over very fine scales. The variation in the mean temperature is similar in all groups. Over the equator strong semi-annual variation is observed (with maximum during November and May) and over mid and high latitudes strong annual variation is observed with maximum during summer and minimum during winter. The sudden stratospheric warming (SSW) of 2010 is also observed in the northern hemisphere during January - February. The migrating diurnal tide, DW1, is very prominent at 30 km over the equatorial region with amplitudes in the range 1-1.5K and showing a band like structure around the equator. The band is slightly shifted towards winter poles, i.e., northward during northern hemisphere winter and southward during southern hemisphere winter. Over latitudes greater than 45° in the winter hemisphere intermittent patches of DW1 are observed. The amplitudes during January 2010 are particularly larger and in the range of 2-3 K which coincide with the occurrence of the SSW of 2010. The SPW1 shows very large amplitudes, reaching 18 K, over mid-latitudes beyond 45° in the winter hemisphere. Specifically, the amplitude of this wave is stronger in the northern hemisphere than in the southern hemisphere. Further, there is a modulation in the amplitude of this wave by, what seems to be, a planetary wave of period ~60 days. These plots show that all groups





give similar results and thereby, for the rest of the paper analysis from group G0 (that considers data from all six satellites) is only discussed.

Results obtained using data over ±30 days from one satellite C004 are very different, particularly in the mid and high latitudes. The mean temperature is smoothened over the 60 day period and the difference between mean temperature of
group G0 and C004 in the last column shows periodic variations of ~60 days over the entire global region. The differences are maximum during winters in the high latitudes and are beyond ± 3K. The amplitude of DW1 over the equator and low latitudes upto 30° are similar to those obtained from the analysis of the groups, however, the values are unusually large over mid and high latitudes, particularly over the regions poleward of 45°. The differences show that DW1 amplitudes from C004 using ±30 days data are overestimated by more than 6K, which is significant, given that the maximum amplitudes of DW1
(from the groups analysis) in high latitudes is less than 3 K. The amplitude of SPW1 is, on the other hand, similar to the variation observed in the analysis of the data in groups, however, the former is smoothened out over the time duration considered. Here also, the difference panel in the last column shows that the SPW1 amplitude observed by data from C004 alone is modulated by periodic variations of ~60 days, particularly in the high latitude winter atmosphere. These differences are also of the order of ±3 K, which are small compared to the maximum SPW amplitudes.

Figure 3 shows the annual mean of the various wave parameters of the current study using group G0 and satellite C004. The annual mean of mean temperature is similar in both columns. However, the migrating diurnal tide as well as non-migrating tides are overestimated by C004, particularly in the high latitudes. Over equator and low latitudes, the tidal amplitudes are similar. Further, SPW1 amplitudes are marginally underestimated by C004. This could be due to the effect of smoothening as more data was used.

Figure 4 shows the variation of amplitudes and phases of DS0 (left column), DW1 (middle column) and DW2 (right column) during the winter of 2009/10, i.e., from December 2009 to February 2010 over 65°N in the top two and over equator in the bottom two rows, respectively. These results are obtained from group G0. Over high latitudes in the winter hemisphere, DW1 shows large amplitudes of 2K, but only intermittently. DS0 and DW2 also show similar intermittent behaviour over winter hemisphere in the high latitudes with amplitudes varying in the range 1-2 K. The phase plots of these tides do not
show any specific pattern as the waves themselves are intermittent. Over the equator, amplitude of DW1 is maximum at 30 km and is in the range of 1-1.5 K. Small amplitudes of 0.5 to 1 K are observed for DS0 and DW2 on either side of this equatorial band at 35 km, however, at all other altitudes over the equator and low latitudes their amplitudes are zero (not shown here). The DW1 maximum at 30 km over equator peaks at noon indicating that the observed tidal variation is due to ozone absorption and is in phase with the sun. The altitude variation shows that the vertical wavelength of DW1 is ~25 km as
known from previous studies.

Figure 5 shows the variation of amplitude and phase of SPW1 at various altitudes from 20 to 50 km (along the different rows). Large amplitudes of SPW1 are seen in the high latitude winter atmosphere and amplitudes of SPW1 over northern hemisphere are largest during winter, with values reaching 18 K at 30 km altitude. It is also seen that in January 2010, as the SSW started amplitude of SPW1 reduced to almost 6K. In the southern hemisphere the amplitudes are smaller and show a





maximum of ~10 K at 30 km. Importantly, there is significant variability over periodicities of ~60 days at all altitudes, with no phase change, in both hemispheres. Investigation of the phase plots show that the phase lines are nearly constant when large amplitudes of the SPWs are observed, and phase lines at each altitude show that the vertical wavelength of SPW1 is in the range from 50 to 60 km at ~60° N, as well as ~60° S.

Comparison of Figures 4 and 5 shows that DS0 and DW2 occur at the time of occurrence of SPW1, however, there does not

seem to be any significant correlation between the non migrating tides and the stationary planetary wave. To investigate this aspect further, Figure 6 shows mean temperature in black, amplitudes of the DS0 in blue, DW1 in red, DW2 in magenta, and SPW1 in green at 30 km at 65° N, equator and 65° S, in the three panels, respectively. The amplitudes of the waves are indicated by the axis on the right of the figure and that of SPW1 is scaled down by a factor of 10 for convenience. The most striking feature of the figure is the occurrence of SSW in January 2010 at 65° N. Exactly during the time when the

temperatures are increasing at a high rate, the amplitude of SSW reduced drastically and the amplitudes of the three tides increased. Amplitude of DS0 is almost 2 K, that if DW1 is 1.5 K and that of DW2 is 1 K. Similar peaks are also observed after the event when the temperature is reducing. At this point, DW1 shows maximum amplitude of 2 K, DS0 shows amplitude of 1 K and DW2 shows amplitudes smaller than 1K. During summer there is no wave activity in both hemispheres. In the southern hemisphere at 65° S, tidal activity is observed as the temperatures start to rise in the winter.

Particularly, as the mean temperature increases at a fast rate during July 2010, amplitude of DW1 is highest and about 2K. DS0 and DW2 are of the order of 1 K. During the next 3-4 months, intermittent patches of DW1 and DW2 of amplitudes ~1.5 K are observed. Over the equator, on the other hand, the picture is very simple as there is no occurrence of DS0 and DW2. Interestingly, DW1 shows significant short term variability of periodicities of the order of 30 days. If data were analysed over 60 days, this variability would not have been observed. Further, the amplitudes of DW1 are marginally higher

(~ 1.5 K) during northern hemisphere winter and are smaller (~1K) during summer.

To understand the simultaneous occurrence of non migrating tides and stationary planetary waves, a simple correlation study is performed and is shown in Figure 7. The first panel shows the correlation of DS0 and SPW1 at 65° N at 30 km for winter from December 2009 to February 2010.  It may be noted that there is no correlation at all. The second panel shows the correlation between DW2 and SPW1 over the same latitude and altitude during June to August 2010. Here also the

correlation is not significant. The bottom panels show similar correlations for the southern hemisphere and there is good correlation between DS0 and SPW1 and a reasonable correlation between DW2 and SPW1. However, the amplitudes of the tides are all almost smaller than 1 K. Thus no statistical relation can be established between occurrence of non migrating tides and stationary planetary waves.

Figure 8 shows another correlation study between mean temperature and DW1 tide. It may be observed that there is no

significant correlation between the two either in the northern or the southern hemisphere. The panels on the right show the variation of DW1 as a function of gradient in the mean temperature. It is clearly noted that when the latter are larger than ±0.25 K day⁻¹, the amplitudes of DW1 are also very large and increasing with increasing gradient. The situation is same in the southern hemisphere too. This effect is not observed when the gradients are smaller and then the amplitudes of DW1 are





smaller than 0.5 K, and are negligible. This clearly indicates an aliasing of energy due to varying mean temperature into the

DW1 tide amplitude.

## 5 Aliasing

It was very clearly established that varying mean temperatures alias into the DW1 tide using SABER data [*Sakazaki et al., 2012*]. However, in the case of COSMIC data, as the data sampling is irregular, it is difficult to establish such aliasing phenomena. To circumvent this problem, numerical experiments are performed to understand the amount of aliasing

occurring with respect to COSMIC data sampling. For the times and locations of COSMIC measurements, a fake numerical atmosphere is created that consists of known variabilities. Table 1 describes 10 such cases considered for this study. The results from these numerical experiments are shown in Figures 9 and 10 and are explained in detail in the table.

Table 1: Numerical Experiments to investigate the aliasing of energy from one component to another as seen in Figures 9 and 10

| S. No | Atmosphere | Equation | Remarks |
|---|---|---|---|
| 1 | Constant SPW1 Amplitude | $T = 10 * \cos(2\pi\lambda/360)$ | SPW1 amplitude extracted with no errors over both latitudes |
| 2 | Slowly varying SPW1 amplitude | $T = 10 + [10 * \cos(2\pi t/50)] * \cos(2\pi\lambda/360)$ | Maximum amplitude is underestimated and minimum amplitude is overestimated. No Aliasing is observed over equator, however, over 65N, DS0 and DW2 components show aliasing of 1 K amplitudes. This happens at times of maximum gradient in SPW1 amplitude. |
| 3 | Fast Varying SPW1 amplitude | $T = 10 + [10 * \cos(2\pi t/10)] * \cos(2\pi\lambda/360)$ | Average SPW1 amplitude is extracted. The periodic variation of 10 days is lost in the analysis. However, no DS0 and DW2 components are observed over both latitudes. |
| 4 | Constant DS0 Amplitude | $T = 10 * \cos(2\pi h/24.)$ | Constant DS0 component extracted successfully. No aliasing into any other component is observed over both latitudes. |
| 5 | Varying and large DS0 Amplitude | $T = 10 + [10 * \cos(2\pi t/10)] * \cos(2\pi h/24.)$ | Average DS0 component is extracted. And no aliasing into other components is seen over both latitudes. |





| 6 | Varying and Small DS0 Amplitude | $T = 1 + [1 * \cos(2\pi t/5)] * \cos(2\pi h/24.)$ | Average DS0 component is extracted. And no aliasing into other components is seen over both latitudes. |
|---|---|---|---|
| 7 | Varying and Small DW2 Amplitude | $T = 1 + [1 * \cos(2\pi t/5)] * \cos(\frac{2\pi h}{24.} + \frac{2\pi \lambda}{360} 2)$ | Average DW2 component is extracted. And no aliasing into other components is seen over both latitudes. |
| 8 | Fast Varying Mean Temperature | $T = 280 + 10 * \cos(2\pi t/10.)$ | Average mean variation is extracted. Aliasing is observed in DW1, SPW1 and DS0 and DW2, all showing amplitudes upto 1 K over equator and larger over 65°N. The large peaks observed in DW1 and SPW1 over 65°N indicate significant aliasing. |
| 9 | Slowly Varying Mean Temperature | $T = 280 + 10 * \cos(2\pi t/50.)$ <br><br> (Similar to variation during SSW 2010) | Mean variation is extracted reasonably (in the range 273 to 287 K), with maximum amplitudes underestimated by 3 K and minimum values overestimated by 3 K. Aliasing is observed in DW1, SPW1 and DS0 and DW2, all showing amplitudes upto 1 K over equator and much larger over 65°N. Large peaks of 5 K are observed in DW1 and that of 3 K are observed in SPW1 over 65°N. Interestingly, the peaks in DW1 occur at times of maximum gradient in mean temperature |
| 10 | Constant Mean | $T = 280$ | Mean temperature is extracted and absolutely no aliasing is observed over both latitudes. |


## 6 Discussion and Concluding Remarks

Extraction of tidal variability from satellite measurements with good accuracy with no aliasing is a challenge. Using SABER temperature data, wave characteristics can be extracted over 60 days. The amplitude of SPWs using SABER data are much smaller [Xu et al., 2014] than that obtained in the current study. During northern hemisphere winter, the maximum average
amplitudes from SABER were 7.2±1.02 K at 45° N and 45 km. There was strong temporal correlation between the occurrence of SPWs and the non migrating tides, which led to the conclusion that the latter were produced due to non linear interactions of SPWs and migrating diurnal tides [Xu et al., 2014]. This study concentrated explicitly upon the generation of





these non-migrating tides and hence the conclusions. However, the current study shows that the amplitudes of SPW1 is very large of the order of 18 K and the strong temporal correlation with DS0 and DW2 could also be caused due to aliasing of the

SPW wave into the non migrating tides. Through the numerical experiments, it is observed that the aliasing of SPW1 into DS0 and DW2 is equal and uniform and thus in the actual analysis if DS0 and DW2 are found to be equal and uniform, it can be concluded that they might be due to aliasing. Thus the question of non-linear interactions between SPW1 and DW1 producing DS0 and DW2 is still debatable. Although non linear interactions cannot be entirely ruled out, the current study shows that the contribution of this mechanism in producing non-migrating tides is not as important as indicated by studies

earlier, that are particularly dependent on analysis of SABER data [*Xu et al., 2014*]. In the current study it is seen very clearly that the DS0 and DW2 components are much smaller than those observed earlier using SABER data.

Aliasing problems involved in SABER data are difficult to verify due to lack of similar global observations but comparisons are made with models and reanalysis and it was observed that there are significant inconsistencies in the comparisons between them [*Sakazaki et al., 2018*]. It was found that the amplitude of trapped diurnal migrating tide in the upper

stratosphere is significantly smaller in reanalyses than that in SABER. The current study also indicates that SABER tidal amplitudes are overestimated, particularly in the mid and high latitudes. Results from space time spectral analysis of gridded monthly COSMIC data for the period from 2007 to 2008 also had showed that the DW1 peaks at 30 km over the equatorial latitudes [*Pirscher et al., 2010*]. It was showed in this paper that sampling was insufficient northward of 50° and the spectral amplitude of sampling error was large. However, in the current study we see that the universal time-longitude space is

sufficiently sampled (Figure 1) and by using the least squares method over shorter lengths of data, it is possible to extract the different wave components. The numerical experiments show that with the given sampling and the technique used, it can be verified if the extracted spectral components SPW1, DS0, DW1, DW2 are geophysical or are a result of aliasing.

During the SSW of 2010, the peaks observed in DW1 and DS0 & DW2 seem to be arising out of aliasing. At 65° N, as the temperature increases (decreases) steeply during the onset (decay) of the warming episode, the DW1 component is observed

to be large (1.5 to 2 K). The entire SSW event lasted over ~60 days approximately and is very similar to the numerical experiment in Case 9, where significant aliasing into DW1 is observed. This experiment clearly shows that over high latitudes, when there is a large gradient in the mean temperature, peaks of large amplitude of DW1 are observed, which are not geophysical in nature. At the same time, the SPW1 component steeply reduced during onset of the episode, due to which the DS0 and DW2 components might have arisen. Here also, there is a contribution of aliasing to SPW1 of the order of 2-3

K, but in reality the observed SPW1 amplitudes are much larger (18 K). Further, the amplitude of 2 K of DS0 during the onset of the event might have some geophysical meaning associated with it, but needs further investigation.

Short term tidal variability was deduced by using data from Sodium Lidar and simultaneous SABER retrievals and TIME-GCM results in the mesosphere and lower thermosphere [*Liu et al., 2007*]. They found large tidal variability which could be the result of interactions with the planetary waves. The migrating diurnal tidal amplitude was modulated by the planetary

wave of 5-7 day period. Such interactions are worth studying in the future using COSMIC data by considering travelling planetary waves to obtain more insights into the tidal variability. Unfortunately, the altitude coverage of COSMIC is only up





to the stratopause and thus tidal characteristics cannot be extracted for altitudes above 45 to 50 km. However, the current study clearly establishes the fact that with COSMIC data short term tidal variability can be obtained with significant understanding of the aliasing involved. The following may thus be concluded from the current study.

1.  COSMIC data is better suited for tidal studies due to better phase sampling of tides and waves; however, due to lack of altitude coverage the studies are confined only to the stratosphere.

2.  The migrating diurnal tide (DW1) is found to be maximum at 30 km over the equator and is attributed to ozone absorption. The vertical wavelength is of the order of 25 km.

3.  Stationary planetary wave of wave number one (SPW1) peaks in the winter hemisphere over high latitudes. It exhibits

strong ~60 day variability with a vertical wavelength of 50 to 60 km at 65° N. This variability could not be observed earlier in SABER studies.

4.  DS0 and DW2 components are relatively small and only observed intermittently in the high latitude middle atmosphere. Most of the peaks seem to be appearing due to aliasing.

5.  Aliasing is significantly reduced when data is analysed over ±10 days using COSMIC data. However, it still exists and

the numerical experiments performed in the current study show that DS0 and DW2 components arise when there is a steep gradient in the SPW1 amplitude over time. Similar aliasing into DW1 component is prominently observed when there is a steep gradient in the mean temperature, particularly in the high latitudes.

6.  These exercises indicate that at the time of the SSW in January 2010, the peaks observed in DW1, as well as DS0 and DW2, could be a manifestation of the aliasing effects involved in satellite data analysis, and that they may not be

geophysical. Thus analysis of satellite data needs to be done extremely carefully in identifying the various tidal components and their characteristics.

It is thus concluded that non linear interactions are not a very important source of generation of non-migrating tides in the winter high latitude stratosphere.

**Code Availability**

The codes are prepared in IDL and can be supplied on request.

**Data availability**

Data used in the current study is obtained from UCAR/COSMIC. The data is freely available.





**Author contribution**

UD and WW conceived the idea.

UD performed the data analysis.

CJP provided insights into usage of COSMIC data.

WW designed and SKD performed the numerical experiments.

UD and WW analysed and finalised the results after discussion with all authors.

UD prepared the manuscript with contribution from all authors.

**Competing interests**

The authors declare that they have no conflict of interest.

**Acknowledgements**

Authors acknowledge UCAR/COSMIC program for providing free access to FORMOSAT-3/COSMIC 'atmPrf' temperature

data. UD is supported by Early Career Research Award ECR/2017/002258 by Science and Engineering Board (SERB), Govt. of India. CJP is supported by the Ministry of Science and Technology of Taiwan through the grant MOST-107-2111-M-008-006.



**Figures**

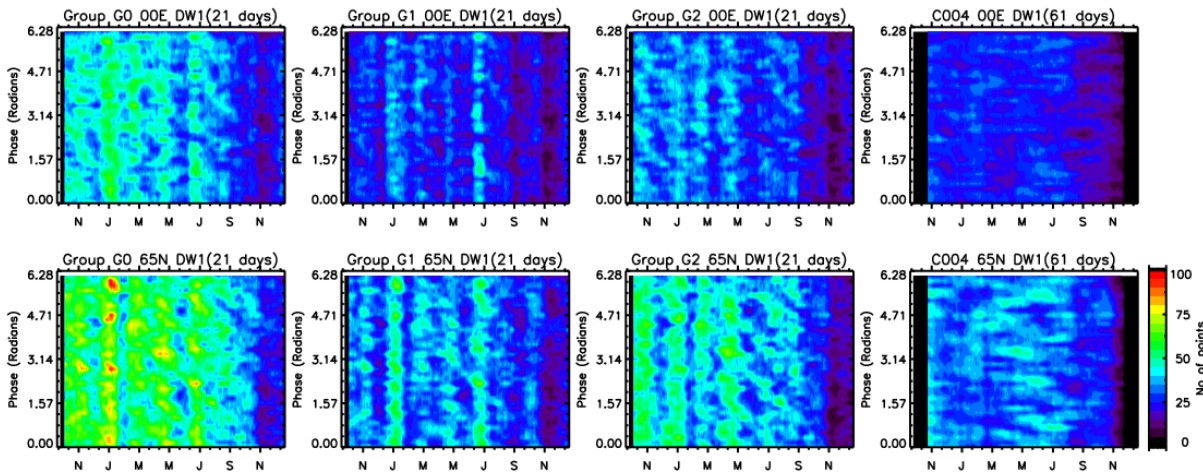

Fig 1.    Distribution of Phase space for the tide DW1 from groups G0, G1, G2  (±10 days' data) and for the satellite C004
(±30 days' data) during the study period (2009-2010) over the equator and 65°N.


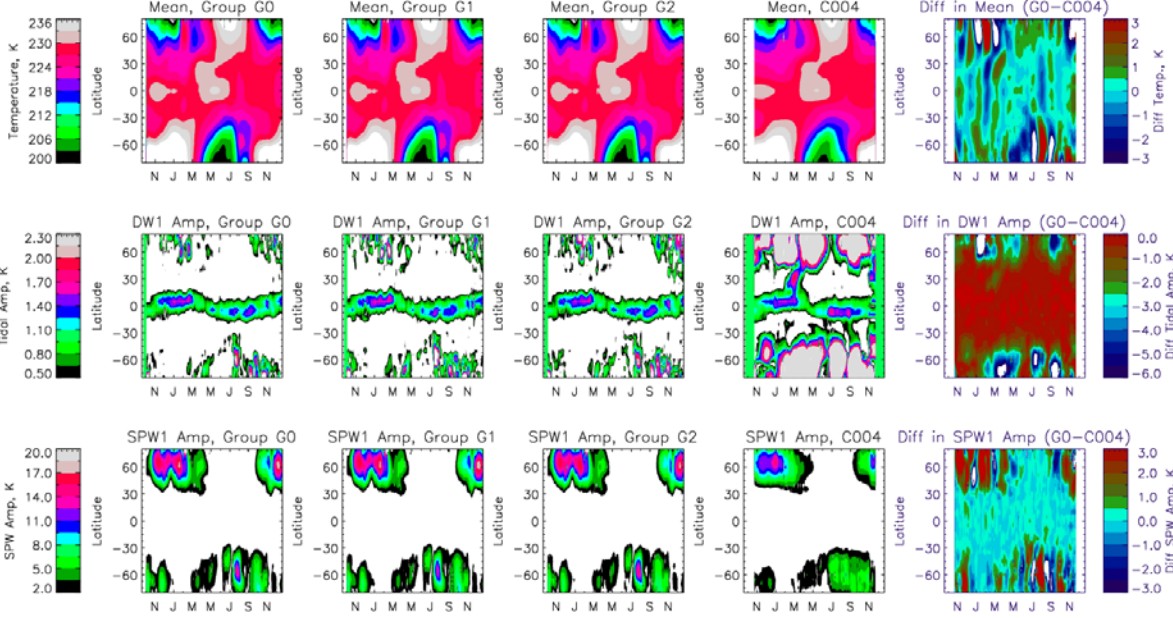

Fig 2.    Rows show the variation in mean temperature, amplitudes of the DW1 tide and SPW1 wave, respectively. Columns
are results from groups G0, G1 and G2, and from satellite C004, respectively, for 30 km altitude. The last column is
the numerical difference in results obtained from group G0 (±10 days) and C004 (±30 days).



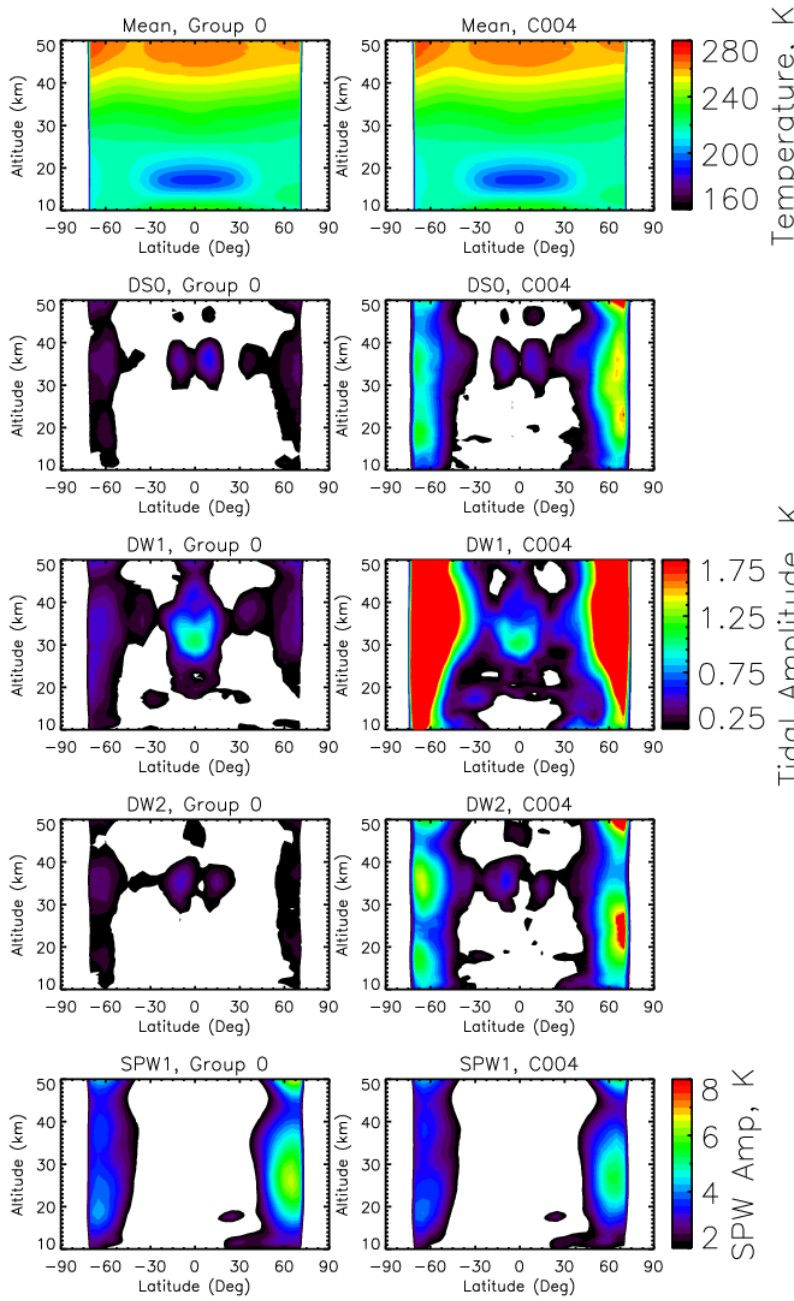

Fig 3.    Annual means of mean temperature ($T_o$) and amplitudes of diurnal tides (DS0, DW1, DW2) and stationary planetary wave (SPW1) for group G0 and satellite C004 during 2010 (Jan to Dec). Note the overestimation of amplitudes of the migrating tide, particularly in mid and high latitudes in the analysis of data over ±30 days using a single satellite (C004).



Fig 4.    Variation of amplitudes and phases of DS0 (left column), DW1 (middle column) and DW2 (right column) during the winter of 2009/10, i.e., from December 2009 to February 2010 over 65°N and equator in the top two and the bottom two rows, respectively.




Fig 5.    Variation of amplitude and phase of SPW1 at various altitudes from 20 to 50 km.



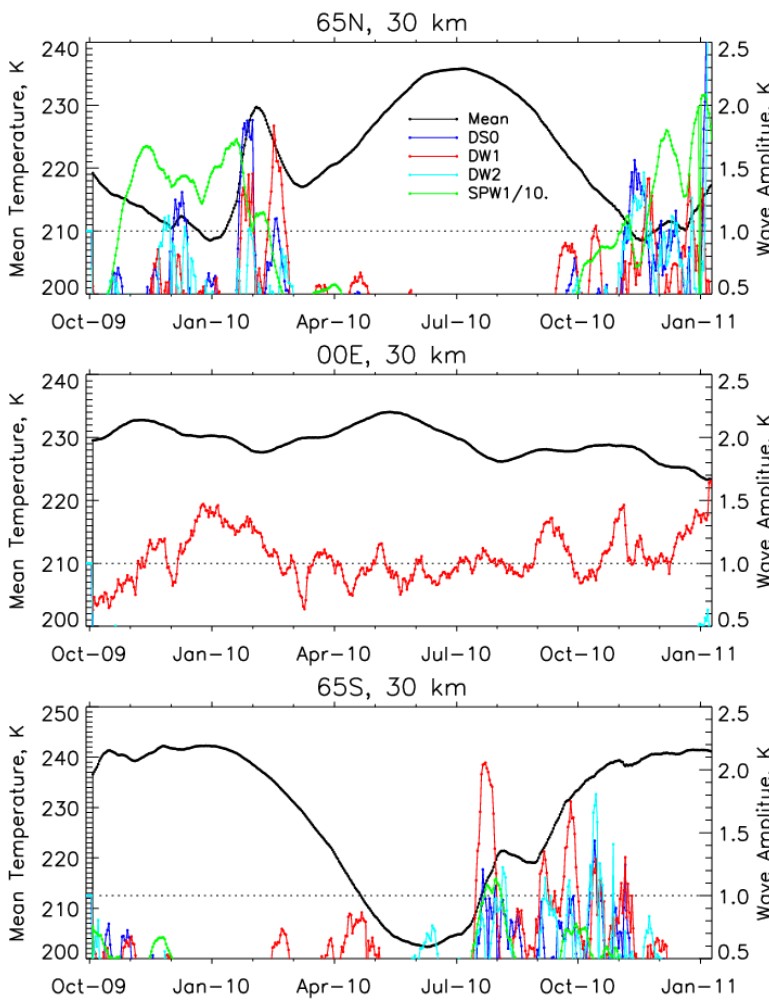

Fig 6. Mean temperature (black), amplitudes of the DS0 (blue), DW1 (red), DW2 (cyan), and SPW1 (green) at 30 km at 65°N, equator, and 65°S during the study period.





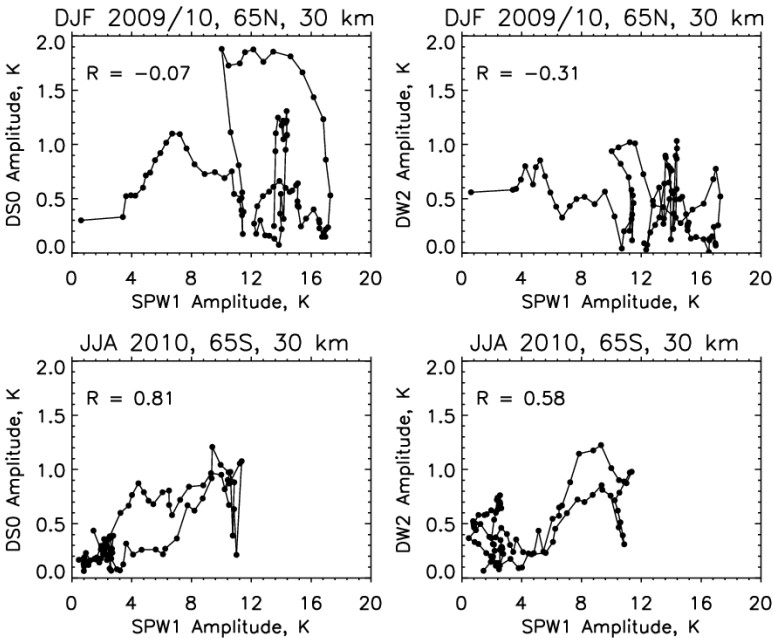

Fig 7.    Correlation between DS0 & SPW1 and DW2 & SPW1 during winters at 65° latitude in the northern and southern
hemispheres.

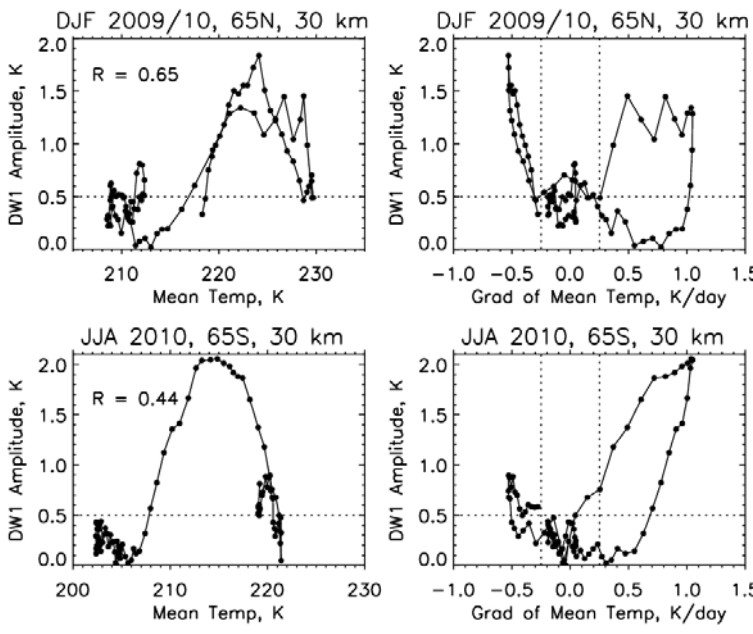


Fig 8.    Correlation between DW1 & mean temperature and DW1 & gradient in mean temperature during winters at 65°
latitude in the northern and southern hemispheres.







Fig 9.    Results of numerical experiments from cases 1 to 7 (Table 1) for atmospheres considered to have only   one
variability among SPW1, DS0 and DW2.





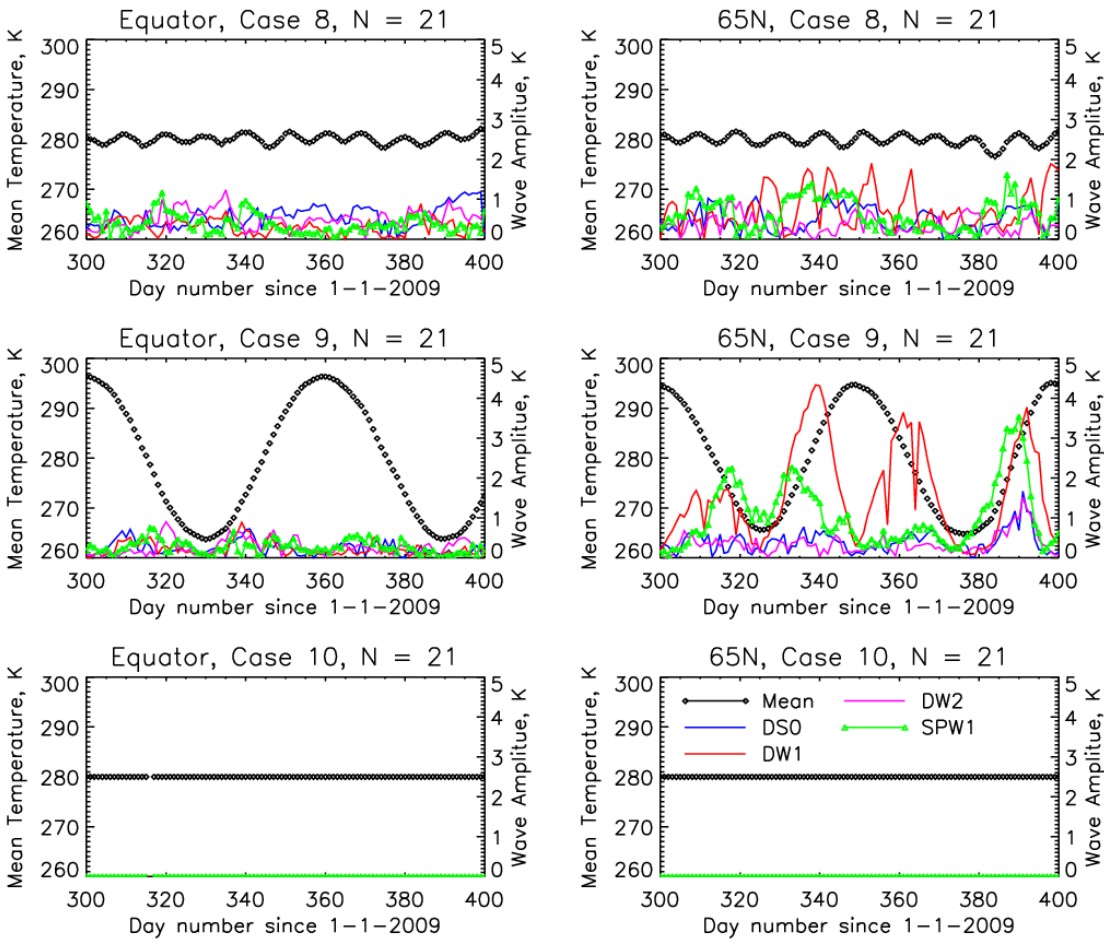

Fig 10. Results of numerical experiments from cases 8 to 10 (Table 1) for atmospheres considered to have only a mean temperature variation.

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
