# Peer review of "Migrating and Non-Migrating Tides Observed in the Stratosphere from FORMOSAT-3/COSMIC Temperature Retrievals"

_Annales Geophysicae, 2019_

## Referee Comment (RC1) · Anonymous Referee #1 · 24 Oct 2019

The paper by Uma Das et al. presents a tidal analysis based on COSMIC temperature data derived during the years 2009 and 2010. The analysis uses the least-square fitting technique concerning the zonal wave number, the longitude, and considering a mean, a diurnal, semidiurnal and terdiurnal tidal as well as a stationary planetary wave component. The data is divided into two groups containing four micro satellites each with a third group containing all 6 satellites as comparison. The present study focuses on the investigation of different lengths of data sets and thus, their influence on each tidal component according to their generation, variability and potentially produced aliasing effects. The findings are relevant especially for the satellite community because they give evidence to new interpretation possibilities regarding the nature of

tides in the Earth's atmosphere.

The paper is well written, the methods are adequately described and the results are properly discussed, therefore a publication in Annales Geophysicae is highly recommended. But nevertheless, some major and minor concerns should be addressed as indicated below before publishing.

Specific comments:

1) In the introduction the authors mentioned that a short-term variability from satellite data cannot be retrieved due to the local time coverage at one particular location, which takes several tens of days (Page 2, line 33-34). In general this statement is true, but nevertheless, there are attempts to derive a short-term variability using the deconvolution method for non-migrating tides or by combining satellite data with ground-based measurements. For a comprehensive overview please add some information about this in the introductory part (for this see e.g.: Oberheide et al., 2002; Pedatella et al., 2016).

2) The classification of the different groups (G1 and G2) seems to be somehow oversampled because 50% of the data is in both groups. The similarity of the results from the different groups is not really surprising when almost the same data is used for each group (G0 contains all six data sets). Perhaps, it is worth to add some information about the exact amount of data from each satellite in form of a table or something like that?

3) Page 4, line 96: What is meant with the statement: "This non-uniform and patternless sampling is in a way supportive to the current study"? Even if there would be a pattern sampling, by knowing this, it would be also possible to distinguish between such a pattern and a real geophysical variation of a tidal signal. Please clarify the meaning with some arguments.

4) Page 4, line 126: "the phase sampling is uniform on any given day for all waves".

The reviewer does not see this in Fig. 1 as just the phase of the DW1 component is shown here. In addition to this, perhaps misleading, statement, it seems to be a contradiction to the comment in the beginning about the non-uniform sampling (see point 3). Uniform suggests a complete sampling for each season. This seems to be not true, especially for the winter months. Please rethink about both statements and rewrite them.

5) Fig. 2: Why there is no amplitude signal for the DW1 tide visible between 15 and 45° N/S? From the theoretical point of view, one would assume that there is a strong signal at the equator which slowly decreases towards the pole. Are the values really zero here or are they not determined? This is not clear from the colormap of the colorbar.

6) Page 7, line 190/191: "with no phase change". This reviewer is not convinced about this statement. From Fig. 5 it is clearly visible that the phase is changing between maxima of amplitudes.

7) The nonlinear interaction of the SPW1 and the migrating diurnal tide as a potential generation mechanism of the DW2 and DS0 tidal component was also investigated by Lieberman et al., 2015. Although this study is focused in the MLT region, it is worth to discuss the results here (page 9, line 241/242) because the authors also used SABER data and derived a day-to-day variability of the tidal components. They also made a numerical investigation of their correlation and found that the interaction is a potential source of the non-migrating tidal components. Their findings contradict an aliasing effect as origin for the non-migrating tidal components. Please enhance the discussion according to this.

8) COSMIC data show significant tidal variabilities over times less than a month. Unfortunately, the data is limited to stratospheric altitudes, and therefore a comparison to other data sets can hardly be made as they are often done for the MLT region due to larger tidal amplitudes at these altitudes. However, there are a few studies which investigate the tidal variability also in the stratosphere, which should also be mentioned

in the discussion (e.g., Baumgarten et al., 2018, Baumgarten and Stober et al., 2019). Both studies show a huge variability of the diurnal tide which origin was not finally determined, but nevertheless, they used also global tidal fields to distinguish between migrating and non-migrating tides. Their results also show a negligible non-migrating tidal contribution if an intermittency is allowed in the analysis which supports a potential aliasing effect as origin for the non-migrating tidal components.

Minor/Technical comments:

- Page 1, line 10: sun → Sun

- Citations should follow the Ann. Geophysicae standard, so please use the command \citet or \citep if latex is used or "()" instead of "[]" for references.

- Please add a short summary of the structure of the manuscript at the end of the introduction. For a reader it is much better if a statements about what follows is given there.

- This reviewer suggests to integrate Section 2.1 into Section 2 or Section 2.1 should get an own Section number. In general, it is not likely to write a subsection if just one subsection is existing.

- Page 4, line 109: in to → into

- Page 4, line 123: What is meant by "viz."? Please clarify.

- Page 4, line 124: "the total phase of the important wave component DW1 are investigated" must be "is investigated"

- Page 4, line 126: "data points reduced" should probably be "data points are reduced"

- Page 5, line 1: "the period investigated" please clarify which period this is (Nov. 2009-Sept. 2010) as this should be stated somewhere here in the data section.

- Page 5, section 3: To be complete here an explanation for the different variables is

needed for the reader.

- Page 5, line 144: How is "with minute differences" determined? The results show small differences between the three groups, but they are not in the range of minutes. If it is meant that the flightpath of the six micro satellites have a temporal difference of a few minutes, then it becomes not clear from the statement written in the manuscript.

- Page 5, line 145: "strong semi-annual variation is observed" The statement should be weakened. The semi-annual variation is visible, but the annual variation is much stronger (the difference here is larger than 30 K, while the difference for the semi-annual variation is less than 10 K).

- Page 5, line 149: Just a thought (there is probably no action required): Is it possible that the band like structure around the equator has something to do with the geomagnetic equator because there is a more or less anti-correlation between both?

- Page 6, line 161: The strongest differences occur not only during winter, also during autumn.

- Page 6, line 172: "are overestimated by C004, particularly in the high latitudes". This statement is not quite correct, the overestimation occurs only in high latitudes, not above the equator. Please modify the sentence.

- Page 6, line 173: "due to the effect of smoothening as more data was used" It is not clear why this is the explanation. Does C004 contain more data than the other satellites?

- Page 6, line 184: Could the authors estimate also the vertical wavelength of the other two tidal components? Probably this is not easy, but at least some sentence about the relation to other tidal components would be helpful.

- Page 7, line 200: The amplitude of the SSW is not seen in the figure, therefore the formulation should be changed.

[Figure]

- Page 10, line 257 and 258: "Showed" is the past tense of "show", but here the authors have to write the past participle which is "shown".

References:

Baumgarten, K., Gerding, M., Baumgarten, G., and Lübken, F.-J. (2018), Temporal variability of tidal and gravity waves during a record long 10-day continuous lidar sounding, Atmos. Chem. Phys., 18, 371–384, https://doi.org/10.5194/acp-18-371-2018.

Baumgarten, K. and Stober, G. (2019), On the evaluation of the phase relation between temperature and wind tides based on ground-based measurements and reanalysis data in the middle atmosphere, Ann. Geophys., 37, 581–602, https://doi.org/10.5194/angeo-37-581-2019.

Oberheide, J., Hagan, M. E., Roble, R. G., and Offermann, D. (2002), Sources of nonmigrating tides in the tropical middle atmosphere, J. Geophys. Res., 107(D21), 4567, doi:10.1029/2002JD002220.

Pedatella, N. M., Oberheide, J., Sutton, E. K., Liu, H.‐L., Anderson, J. L., and Raeder, K. ( 2016), Short‐term nonmigrating tide variability in the mesosphere, thermosphere, and ionosphere, J. Geophys. Res. Space Physics, 121, 3621– 3633, doi:10.1002/2016JA022528.

Lieberman, R. S., Riggin, D. M., Ortland, D. A., Oberheide, J., and Siskind, D. E. ( 2015), Global observations and modeling of nonmigrating diurnal tides generated by tide‐planetary wave interactions, J. Geophys. Res. Atmos., 120, 11,419– 11,437, doi:10.1002/2015JD023739.

---

## Referee Comment (RC2) · Anonymous Referee #2 · 28 Oct 2019

General comments

Temperature data from the FORMOSA Satellite Series No. 3/Constellation Observing System for Meteorology, Ionosphere and Climate (FORMOSAT-3/COSMIC) observations obtained during 2009 to 2010 have been used to analyze migrating (DW1) and non-migrating (DS0 and DW2) diurnal tides in the middle atmosphere from 10 to 50 km over equator and 65°N. For this, the authors have separated two overlapping groups with data from 4 satellites each besides to consider data from 6 satellites group. The analysis of each group was performed considering 21 days data centred over each 11 days. The topic of the manuscript is interesting for understand the source of generation

of non-migrating tides in the high latitude during winter, as well as to identify aliasing effects in satellite data analysis. The manuscript presentation is clear and the scientific contribution is appropriate for this journal. However, there are some moderated issues that need to be addressed.

Specific comments

Some important works that deal with short-term variability of the tides were not contemplated. For example: using NAVGEM-HA reanalysis and meteor radars McCormack et al. (2017) have observed day-to-day variability of the winds and tides. Recently, Baumgarten and Stober (2019) have estimated the tidal variability from 10-day continuous lidar observation.

Information about the COSMIC mission and its temperature profiles was not sufficiently addressed. Could the authors include a brief summary containing minimal information about the COSMIC mission as well as the temperature derivation process?

What is the COSMIC post-processed level of data used in the study? The authors could also add information about the spatial distribution of the COSMIC observations (vertical and horizontal resolutions) of the data used in the study.

Based on correlation analysis between tides and SPW1 amplitudes, the authors claim that the contribution of nonlinear interaction to non-migrating tides generation is not important. To provide convincing support for this finding, an effort should be undertaken to include additional analysis (for example, cross-correlation and phase coherence).

The discussion needs to be improved considering some studies on tidal variability in both stratosphere and mesosphere. For example: nonmigrating diurnal tides generated by tide-planetary wave interactions have been studied by Lieberman et al. (2015), and Niu et al. (2018) have discussed this issue and their relationship to SSW.

Minor/Technical comments:

Throughout the manuscript, many acronyms were used without proper designation.

Please provide compound term on first appearance.

Lines 28-29: "TIMED satellite" Line 45: "satellite observations of TIDI and SABER instruments onboard TIMED" Line 46: "UARS"

Line 105: change "As mentioned earlier 10 days data from all six COSMIC" to "As mentioned earlier +-10 days data from all six COSMIC"

References:

McCormack et al. Comparison of mesospheric winds from a high-altitude meteorological analysis system and meteor radar observations during the boreal winters of 2009-2010 and 2012-2013, J. Atmos. Solar-Terr. Phys., 154, 132-166. 2017.

Baumgarten, K. and Stober, G. On the evaluation of the phase relation between temperature and wind tides based on ground-based measurements and reanalysis data in the middle atmosphere, Ann. Geophys., 37, 581–602. 2019.

Lieberman, R.S.; Riggin, D.M.; Ortland, D.A.; Oberheide, J.; Siskind, D.E. Global observations and modeling of nonmigrating diurnal tides generated by tide-planetary wave interactions. J. Geophys. Res. Atmos., 120, 11419–11437. 2015.

Niu, X. Du, J., Zhu, X. Statistics on Nonmigrating Diurnal Tides Generated by Tide-Planetary Wave Interaction and Their Relationship to Sudden Stratospheric Warming. Atmosphere, 9, 416. 2018.

---

## Author Response (AR2)

**Author's Reply to Topical Editor**

Title: Migrating and Non-Migrating Tides Observed in the Stratosphere from FORMOSAT-3/COSMIC Temperature Retrievals

Authors: Uma Das, William E. Ward, Chen Jieh Pan, Sanat Kumar Das

**MS No.: angeo-2019-140**

Authors thank the Topical Editor for encouraging this manuscript.

Authors also thank both Reviewers for their positive comments and suggestions. Point by point replies are given in ensuing pages (# 2 to 12). As detailed, all comments have been addressed and suggestions have been incorporated.

In addition to the changes suggested by Reviewers, authors have completed a thorough checking of the manuscript for grammar mistakes, typographical errors and also included all acknowledgements. All changes in the revised manuscript are marked in track change mode and given in this document in pages # 13 to 37.

We sincerely hope that the revised manuscript is now clear and concise and suitable for publication in your esteemed journal.

**Reply to Anonymous Referee #1**

(https://doi.org/10.5194/angeo-2019-140-RC1, 2019)

Authors thank the Reviewer for her/his positive comments and suggestions. Please note that point by point replies are given below in blue. We sincerely hope that the revised manuscript is now clear and concise.

The paper by Uma Das et al. presents a tidal analysis based on COSMIC temperature data derived during the years 2009 and 2010. The analysis uses the least-square fitting technique concerning the zonal wave number, the longitude, and considering a mean, a diurnal, semidiurnal and terdiurnal tidal as well as a stationary planetary wave component. The data is divided into two groups containing four micro satellites each with a third group containing all 6 satellites as comparison. The present study focuses on the investigation of different lengths of data sets and thus, their influence on each tidal component according to their generation, variability and potentially produced aliasing effects. The findings are relevant especially for the satellite community because they give evidence to new interpretation possibilities regarding the nature of tides in the Earth's atmosphere.

The paper is well written, the methods are adequately described and the results are properly discussed, therefore a publication in AnnalesGeophysicae is highly recommended (Thank you). But nevertheless, some major and minor concerns should be addressed as indicated below before publishing.

**Specific comments:**

1) In the introduction the authors mentioned that a short-term variability from satellite data cannot be retrieved due to the local time coverage at one particular location, which takes several tens of days (Page 2, line 33-34). In general this statement is true, but nevertheless, there are attempts to derive a short-term variability using the deconvolution method for non-migrating tides or by combining satellite data with ground-based measurements. For a comprehensive overview please add some information about this in the introductory part (for this see e.g.: Oberheide et al., 2002; Pedatella et al.,2016).
Ans: Authors thank the Reviewer for providing these important references. They are now discussed in the revised manuscript in Section 1 (Introduction).

2) The classification of the different groups (G1 and G2) seems to be somehow oversampled because 50% of the data is in both groups. The similarity of the results from the different groups is not really surprising when almost the same data is used for each group (G0 contains all six data sets). Perhaps, it is worth to add some information about the exact amount of data from each satellite in form of a table or something like that?
The following figure shows the data availability of the number of profiles available at various latitudes from each of the COSMIC satellites. This figure will be given in a Supplementary Section of the paper.

[Figure]

*Figure S1: Number of profiles available over equator, 30N, 45N and 65N, from each of the COSMIC satellites C001 to C006. The last panel shows data available from all satellites taken together.*

3) Page 4, line 96: What is meant with the statement: "This non-uniform and patternless sampling is in a way supportive to the current study"? Even if there would be a pattern sampling, by knowing this, it would be also possible to distinguish between such a pattern and a real geophysical variation of a tidal signal. Please clarify the meaning with some arguments.
Ans: The non-uniform and pattern-less spatial and temporal sampling is in a way supportive to the current study to characterise the variability of tides in the middle atmosphere as we have used the method of least squares fitting. So, aliasing effects are reduced.

4) Page 4, line 126: "the phase sampling is uniform on any given day for all waves". The reviewer does not see this in Fig. 1 as just the phase of the DW1 component is shown here. In addition to this, perhaps misleading, statement, it seems to be a contradiction to the comment in the beginning about the non-uniform sampling (see point 3). Uniform suggests a complete sampling for each season. This seems to be not true, especially for the winter months. Please rethink about both statements and rewrite them.

Ans: COSMIC sampling of the atmosphere is irregular in space and time [Please refer to Figure 1 of Pirscher et al., 2010]. The same can be seen in figure below, which shows the sampling in longitude-UT space for one day and 21 days.

[Figure]

Thus, the information in line 96 is correct. These observations, on the other hand sample the different waves at various phase points. We extracted the phase information of each wave as $2\pi ft+2\pi s\lambda$, where $f$ is frequency and $s$ is wave number of the wave, $t$ is time (UT) and $\lambda$ is longitude of observation, which is plotted in Figure 1.

Here we would like to draw the attention of the Reviewer towards an error in the figure. Figure 1 actually shows phase sampling distribution of the DS0 Wave. Title was mistakenly written as DW1. We have corrected the figure to show DW1 phase sampling distribution in the revised manuscript.

We show here the phase sampling distribution of all the three tides.

[Figure]

[Figure]

On inspection it can be seen that, phase on any given day, i.e., a vertical slice of any panel in these plots, is reasonably uniform, as mentioned in line 126.

5) Fig. 2: Why there is no amplitude signal for the DW1 tide visible between 15 and 45_ N/S? From the theoretical point of view, one would assume that there is a strong signal at the equator which slowly decreases towards the pole. Are the values really zero here or are they not determined? This is not clear from the colormap of the colorbar.
Ans: The amplitudes of DW1 between 15 and 45°N/S are indeed negligible and below 0.5 K, as determined from COSMIC temperature data. It may be noted that these results are for 30 km.

This is also verified with published literature. Figure 1 of Sakazaki et al., 2018 (ACP) shows that DW1 has amplitude of 1 K over equator at 30 km and is below 0.6K at other latitudes as seen in satellite and reanalysis datasets (SABER,JRA-55, JRA-55C, JRA-55AMIP, MERRA-2, MERRA, ERA-Interim, and CFSR). DW1 amplitudes of 3 to 3.5 K are observed between 15 and 45 N/S at higher altitudes of 50 km. Figure 3 (First Row, Second Column) of Hagan and Forbes 2003, shows that GSWM amplitude of DW1 is 1 K over equator at 50 km. Hence comparisons with published literature proves the current results.

6) Page 7, line 190/191: "with no phase change". This reviewer is not convinced about this statement. From Fig. 5 it is clearly visible that the phase is changing between maxima of amplitudes.

[Figure]

SPW1, 65N, 30 km

Ans: The phase of SPW1 is varying between 100 and 200 degree longitude between maxima, as seen on the above figure. Thus the phrase is modified as 'with small phase change'.

7) The nonlinear interaction of the SPW1 and the migrating diurnal tide as a potential generation mechanism of the DW2 and DS0 tidal component was also investigated by Lieberman et al., 2015. Although this study is focused in the MLT region, it is worth to discuss the results here (page 9, line 241/242) because the authors also used SABER data and derived a day-to-day variability of the tidal components. They also made a numerical investigation of their correlation and found that the interaction is a potential source of the non-migrating tidal components. Their findings contradict an aliasing effect as origin for the non-migrating tidal components. Please enhance the discussion according to this.

Ans: Authors thank the Reviewer for suggesting this reference. It is discussed in the revised manuscript.

"Lieberman et al (2015) discuss DW2 variation over equatorial mesopause region and the mechanism of generation is proposed that stratospheric SPWs over mid and high latitudes are ducted upward and equatorward that interact with equatorial DW1 and thereby generate DW2 over the mesoapuse region. DS0 is not quite discussed by Lieberman et al (2015)."

8) COSMIC data show significant tidal variabilities over times less than a month. Unfortunately, the data is limited to stratospheric altitudes, and therefore a comparison to other data sets can hardly be made as they are often done for the MLT region due to larger tidal amplitudes at these altitudes. However, there are a few studies which investigate the tidal variability also in the stratosphere, which should also be mentioned in the discussion (e.g., Baumgarten et al., 2018, Baumgarten and Stober et al., 2019). Both studies show a huge variability of the diurnal tide which origin was not finally determined, but nevertheless, they used also global tidal fields to distinguish between migrating and non-migrating tides. Their results also show a negligible non-migrating tidal contribution if an intermittency is allowed in the analysis which supports a potential aliasing effect as origin for the non-migrating tidal components.

Ans: Authors thank the Reviewer for suggesting this reference. These are discussed in the revised manuscript.

"Baumgarten and Stober (2019) derived short term tidal variability in the altitude range from 30 to 70 km using temperature derived from lidar observations at Kühlungsborn (54°N, 12°E), a mid latitude station. The diurnal tide (consisting of all wavenumbers) in temperature and winds was extracted from lidar data and compared with DW1 component of temperature

and winds from Modern-Era Retrospective analysis for Research and Applications, Version 2 (MERRA-2). It was shown that the local tidal fields are dominated by the migrating diurnal and migrating semidiurnal tides and that other components are negligible. This indicates that the non-migrating components may have very little contribution and thus supports the current study that the observed non migrating tides could be possibly due to aliasing."

**Minor/Technical comments:**

- Page 1, line 10: sun ! Sun
Ans: Corrected

- Citations should follow the Ann. Geophysicae standard, so please use the command ncitet or ncitep if latex is used or "()" instead of "[]" for references.
Ans: Corrected

- Please add a short summary of the structure of the manuscript at the end of the introduction. For a reader it is much better if a statements about what follows is given there.
Ans: Summary added as follows.
"The paper is organised as follows. Section 2 describes the FORMOST-3/COSMIC data used, satellite sampling and phase space of the various wave components. Data analysis method of least square fitting is described briefly in Section 3. Tidal characteristics and associated aliasing are described in Sections 4 & 5, respectively, and the results are discussed and summarised in Section 6."

- This reviewer suggests to integrate Section 2.1 into Section 2 or Section 2.1 should get an own Section number. In general, it is not likely to write a subsection if just one subsection is existing.
Ans: Both sections are combined and renamed as 'Data and Sampling'.

- Page 4, line 109: in to ! into
Ans: Corrected

- Page 4, line 123: What is meant by "viz."? Please clarify.
Ans: 'viz'means 'namely'. It is replaced in the revised manuscript.

- Page 4, line 124: "the total phase of the important wave component DW1 are investigated" must be "is investigated"
Ans: Corrected

- Page 4, line 126: "data points reduced" should probably be "data points are reduced"
Ans: Corrected

- Page 5, line 1: "the period investigated" please clarify which period this is (Nov. 2009-Sept. 2010) as this should be stated somewhere here in the data section.
Ans: The period of study is October 2009 to December 2010. It is specified in the revised manuscript.

- Page 5, section 3: To be complete here an explanation for the different variables is needed for the reader.

Ans: Added

- Page 5, line 144: How is "with minute differences" determined? The results show small differences between the three groups, but they are not in the range of minutes. If it is meant that the flightpath of the six micro satellites have a temporal difference of a few minutes, then it becomes not clear from the statement written in the manuscript.
Ans: 'minute' here means extremely small. To avoid confusion, it is replaced.

- Page 5, line 145: "strong semi-annual variation is observed" The statement should be weakened. The semi-annual variation is visible, but the annual variation is much stronger (the difference here is larger than 30 K, while the difference for the semiannual variation is less than 10 K).
Ans: Modified as suggested.

- Page 5, line 149: Just a thought (there is probably no action required): Is it possible that the band like structure around the equator has something to do with the geomagnetic equator because there is a more or less anti-correlation between both?
Ans: There is no relation with the geomagnetic equator.

- Page 6, line 161: The strongest differences occur not only during winter, also during autumn.
Ans: It is actually during winter and spring. It is corrected in the revised manuscript.

- Page 6, line 172: "are overestimated by C004, particularly in the high latitudes". This statement is not quite correct, the overestimation occurs only in high latitudes, not above the equator. Please modify the sentence.
Ans: Modified as suggested

- Page 6, line 173: "due to the effect of smoothening as more data was used" It is not clear why this is the explanation. Does C004 contain more data than the other satellites?
Ans: Yes, C004 has more data (±30 days), while G0,G1,G2 consider ±10 days data. This is mentioned in Section 3, Analysis.

- Page 6, line 184: Could the authors estimate also the vertical wavelength of the other two tidal components? Probably this is not easy, but at least some sentence about the relation to other tidal components would be helpful.
Ans: From Figure 4, it is difficult it conclusively comment upon the vertical wavelength of the other tidal components, and so we have not added anything in this regard.

- Page 7, line 200: The amplitude of the SSW is not seen in the figure, therefore the formulation should be changed.
Ans: It is a typographical error. It has to be SPW and corrected in the revised version.

- Page 10, line 257 and 258: "Showed" is the past tense of "show", but here the authors have to write the past participle which is "shown".
Ans: Corrected as suggested.

**References:**

Baumgarten, K., Gerding, M., Baumgarten, G., and Lübken, F.-J. (2018), Temporal variability of tidal and gravity waves during a record long 10-day continuous lidar sounding, Atmos. Chem. Phys., 18, 371–384, https://doi.org/10.5194/acp-18-371-2018.

Baumgarten, K. and Stober, G. (2019), On the evaluation of the phase relation between temperature and wind tides based on ground-based measurements and reanalysis data in the middle atmosphere, Ann. Geophys., 37, 581–602, https://doi.org/10.5194/angeo-37-581-2019.

Oberheide, J., Hagan, M. E., Roble, R. G., and Offermann, D. (2002), Sources of nonmigrating tides in the tropical middle atmosphere, J. Geophys. Res., 107(D21), 4567, doi:10.1029/2002JD002220.

Pedatella, N. M., Oberheide, J., Sutton, E. K., Liu, H.â˘ARˇ L., Anderson, J. L., and Raeder, K. ( 2016), Short term nonmigrating tide variability in the mesosphere, thermosphere, and ionosphere, J. Geophys. Res. Space Physics, 121, 3621– 3633, doi:10.1002/2016JA022528.

Lieberman, R. S., Riggin, D. M., Ortland, D. A., Oberheide, J., and Siskind, D. E. (2015), Global observations and modeling of nonmigrating diurnal tides generated by tideâ˘ARˇ planetary wave interactions, J. Geophys. Res. Atmos., 120, 11,419– 11,437, doi:10.1002/2015JD023739.

Ans: Authors thank the Reviewer for providing the above references. They are all now discussed in the revised manuscript.

**Reply to Anonymous Referee #2**

(https://doi.org/10.5194/angeo-2019-140-RC2, 2019)

Authors thank the Reviewer for her/his positive comments and suggestions. Please note that point by point replies are given below in blue. We sincerely hope that the revised manuscript is now clear and concise.

**General comments**

Temperature data from the FORMOSA Satellite Series No. 3/Constellation ObservingSystem for Meteorology, Ionosphere and Climate (FORMOSAT-3/COSMIC) observationsobtained during 2009 to 2010 have been used to analyze migrating (DW1) andnon-migrating (DS0 and DW2) diurnal tides in the middle atmosphere from 10 to 50 kmover equator and 65_N. For this, the authors have separated two overlapping groupswith data from 4 satellites each besides to consider data from 6 satellites group. Theanalysis of each group was performed considering 21 days data centred over each 11days. The topic of the manuscript is interesting for understand the source of generationof non-migrating tides in the high latitude during winter, as well as to identify aliasingeffects in satellite data analysis. The manuscript presentation is clear and the scientificcontribution is appropriate for this journal. However, there are some moderated issuesthat need to be addressed.

**Specific comments**

Some important works that deal with short-term variability of the tides were not contemplated.For example: using NAVGEM-HA reanalysis and meteor radars McCormack etal. (2017) have observed day-to-day variability of the winds and tides. Recently, Baumgartenand Stober (2019) have estimated the tidal variability from 10-day continuouslidar observation.
Ans: Authors thank the Reviewer for providing these important references. They are now discussed in the revised manuscript in Section 6.

"Baumgarten and Stober (2019) derived short term tidal variability in the altitude range from 30 to 70 km using temperature derived from lidar observations at Kühlungsborn (54°N, 12°E), a mid latitude station. The diurnal tide (consisting of all wavenumbers) in temperature and winds was extracted from lidar data and compared with DW1 component of temperature and winds from Modern-Era Retrospective analysis for Research and Applications, Version 2 (MERRA-2). It was shown that the local tidal fields are dominated by the migrating diurnal and migrating semidiurnal tides and that other components are negligible. This indicates that the non-migrating components may have very little contribution and thus supports the current study that the observed non migrating tides could be possibly due to aliasing."

"McCormack et al (2017) investigated the short term tidal variability during the SSWs of January 2010 and January 2013 using high latitude Navy Global Environmental Model (NAVGEM) data in the mesosphere and lower thermosphere region. NAVGEM is a result of assimilation of middle atmospheric data from nine meteor radar stations and other satellite measurements, including those from SABER on board TIMED satellite. Their results show a reduction in semi-diurnal amplitude before the onset of SSW and increases after the event, peaking 10-14 days later."

Information about the COSMIC mission and its temperature profiles was not sufficientlyaddressed. Could the authors include a brief summary containing minimal informationabout the COSMIC mission as well as the temperature derivation process?

Ans: The phase delay of L1 and L2 in signals received is due to change in refractivity which is converted to electron density in the ionosphere and temperature and other parameters in the lower atmosphere and are described in detail in literature (Kuo et al., 2004; Kursinski et al., 1997). Briefly, the Earth's refractive index at microwave wavelengths is affected by the dry neutral atmosphere, water vapour and free electrons in the ionosphere and thus by deriving refractivity of the atmosphere, the above mentioned parameters can be retrieved. This is now included in the revised manuscript.

What is the COSMIC post-processed level of data used in the study? The authorscould also add information about the spatial distribution of the COSMIC observations(vertical and horizontal resolutions) of the data used in the study.

Ans: COSMIC level2 dry temperature 'atmPrf' profiles for lower atmosphere are used in the present study. Data is considered at 1 km intervals from 15 to 50 km. It is known that the vertical resolution of RO derived temperature profiles is 0.5 km in the troposphere and 2 km in the stratosphere (Kursinski et al., 1997, Scherllin-Pirscher et al., 2017). This is now added in the revised manuscript.

Based on correlation analysis between tides and SPW1 amplitudes, the authors claimthat the contribution of nonlinear interaction to non-migrating tides generation is not important.To provide convincing support for this finding, an effort should be undertakento include additional analysis (for example, cross-correlation and phase coherence).

Ans: Figure 7 shows that there is no significant correlation between non migrating tides and SPW1 and thus, no reasonable statistical relation can be established between occurrence of these waves. To investigate this further, the numerical experiments have been performed as described in Table 1 and shown in Figure 9. These numerical experiments show significant aliasing between SPW1 and DS0 & DW2. Based in this latter figure, we are concluding that non-linear interactions may not be as important a source as is believed.

In continuation, and as suggested by the Reviewer we performed a cross correlation study between SPW1 and DS0 & DW2 as a function of lag of ±30 days and is shown below.

[Figure]

This Figure does not add any new information to existing understanding and hence is not included in the revised manuscript.

The discussion needs to be improved considering some studies on tidal variability inboth stratosphere and mesosphere. For example: nonmigrating diurnal tides generatedby tide-planetary wave interactions have been studied by Lieberman et al. (2015), andNiu et al. (2018) have discussed this issue and their relationship to SSW.

Ans: Authors thank the Reviewer for providing these important references. They are now discussed in the revised manuscript in Section 6.

"There are also studies that have shown that time evoltion of DW2 over equatorial mesopause region follows SPW1 variation over high latitude stratosphere (Lieberman et al., 2015; Niu et al., 2018). It is proposed that mid to high latitude stratospheric SPWs are ducted upward and equatorward that interact with equatorial DW1 over mesopause and thereby generate DW2 over the equatorial mesoapuse region. DS0 is not quite discussed by Lieberman et al (2015). Niu et al (2018) invetigated this SPW1-DW1 interaction during SSWs using extended Canadian Middle Atmosphere Model (eCMAM) data and found good but varying correlations during 20 out of 31 SSW events with both DW2 nd DS0 which indicated that the strength of non-linear interactions also varied from year to year. As the correlations are not observed during all SSW events the proposed mechanism of non-linear interactions is still questionable."

**Minor/Technical comments:**
Throughout the manuscript, many acronyms were used without proper designation.

Please provide compound term on first appearance.
Ans: Abbreviations are provided for all acronyms.

Lines 28-29: "TIMED satellite" Line 45: "satellite observations of TIDI and SABERinstruments onboard TIMED" Line 46: "UARS"
Ans: Abbreviations are provided for all acronyms.

Line 105: change "As mentioned earlier 10 days data from all six COSMIC" to "Asmentioned earlier +-10 days data from all six COSMIC"
Ans: It is correctly stated in this line that 10 days of data are in principle sufficient when data from all six COSMIC satellites is considered. If three satellites are considered, 20 days data is required and if only one satellite is considered then 60 days of data is required. In the analysis of the paper ±10 days data is considered for groups G0 (six satellites), G1 (C001, C002, C003, and C004) and G2 (C004, C005, C006 and C001)to maintain uniformity and to avoid data gaps. The text in this paragraph is modified appropriately to clarify this aspect.

**References:**

1. McCormack et al. Comparison of mesospheric winds from a high-altitude meteorologicalanalysis system and meteor radar observations during the boreal winters of2009-2010 and 2012-2013, J. Atmos. Solar-Terr. Phys., 154, 132-166. 2017.
2. Baumgarten, K. and Stober, G. On the evaluation of the phase relation between temperatureand wind tides based on ground-based measurements and reanalysis data in the middle atmosphere, Ann. Geophys., 37, 581–602. 2019.
3. Lieberman, R.S.; Riggin, D.M.; Ortland, D.A.; Oberheide, J.; Siskind, D.E. Globalobservations and modeling of nonmigrating diurnal tides generated by tide-planetarywave interactions. J. Geophys. Res. Atmos., 120, 11419–11437. 2015.
4. Niu, X. Du, J., Zhu, X. Statistics on Nonmigrating Diurnal Tides Generated by Tide-Planetary Wave Interaction and Their Relationship to Sudden Stratospheric Warming.Atmosphere, 9, 416. 2018.

Authors thank the Reviewer for providing the above references. They are all now discussed in the revised manuscript.

**Migrating and  Nonmigrating Tides Observed in the Stratosphere from FORMOSAT-3/COSMIC Temperature Retrievals**

Uma Das[1*], William E. Ward[2], Chen Jeih Pan[3], Sanat Kumar Das[4]

[1]Indian Institute of Information Technology Kalyani, Kalyani, West Bengal, India
[2]Department of Physics, University of New Brunswick, Fredericton, Canada
[3]Institute of Space Science, National Central University, Zhongli, Taiwan
[4]Environmental Sciences Section, Bose Institute, Kolkata, India

*Correspondence to*: Uma Das (uma@iiitkalyani.ac.in)

**Abstract.** Formosa Satellite-3/Constellation Observing System for Meteorology, Ionosphere and Climate (FORMOSAT-3/COSMIC) temperature data during October 2009 to December 2010 are analysed for tides in the middle atmosphere from ~10 to 50 km. COSMIC is a set of six micro satellites in near  Sun synchronous orbits with 30° orbital separations and provides good phase space sampling of tides. Short term tidal variability is deduced by considering ±10 days' data together. The migrating diurnal (DW1) tide is found to peak over the equator at 30 km. It maximises and slightly shifts poleward during winters . Over mid and high latitudes, DW1 and the nonmigrating diurnal tides with wavenumber 0 (DS0) and wavenumber 2 (DW2) are intermittent in nature. Numerical experiments in the current study show that these could be a result of aliasing as they are found to occur at times of steep rise or fall in the mean temperature, particularly during the sudden stratospheric warming (SSW) of 2010. Further, stationary planetary wave component of wavenumber 1 (SPW1) is found to be of very large amplitudes in the northern hemisphere reaching 18 K at 30 km over 65°N. By using data from COSMIC over shorter durations, it is shown that aliasing between stationary planetary wave and nonmigrating tides is reduced and thus results in the large amplitudes of the former. This study clearly indicates that non-linear interactions are not a very important source of generation of the nonmigrating tides in the mid and high latitude winter  stratosphere. There is also a modulation of SPW1 by a ~60 days oscillation in the high latitudes, which was not seen earlier.

**1 Introduction**

Tidal variability in temperature and winds of the atmosphere is a very important parameter to understand the long term as well as day to day variations in the atmosphere.  To date, the nature of short term global tidal variabilities in the middle atmosphere have not been  understood due to lack of sufficient data. Using only ground based data the dominant tidal periods can be identified (She et al 2004, Baumgarten et al, 2018; Baumgarten and Stober, 2019, etc.) but it is difficult to obtain the longitudinal variability (i.e., wavenumber of the tides) unless there are simultaneous measurements at different longitudes along the same latitude circle (Wu et al., 2008). Even if such measurements are possible over a given latitude, all

latitudes of the globe cannot be covered due to various reasons including land-sea distribution. On the other hand, while satellites have the ability to take global measurements their local time coverage is limited. For example, the Thermosphere, Ionosphere, Mesosphere Energetics and Dynamics (TIMED) satellite, which is in a near-Sun synchronous orbit, takes ~60 days to cover all local times at a given location (Mertens et al., 2004; Remsberg et al., 2003; Remsberg et al., 2008). This implies that to derive tidal characteristics, data has to be accumulated for ~60 days (Remsberg et al., 2008; Sakazaki et al., 2012; Xu et al., 2014; Zhang et al., 2006). Even then, due to the satellite's orbit, noon time observations are not available. Thus all phases of the tides, specifically migrating tides, are not sampled. This poses a problem for accurate determination of tidal variabilities. Accumulating data over 60 days also means that the short term variabilities are lost. Further, any changes in the mean variation of the temperature aliases into the energy of migrating tides (Forbes et al., 1997; Sakazaki et al., 2012). A few studies, however, extracted short term tidal varibility using a deconvolution method (Oberheide et al., 2002; Lieberman et al 2015), by combining ground based measurements and reanalysis data with satellite measurements (Pedatella et al., 2016) and using data assimilation models (McCormak et al 2017).

Tides are produced in temperature and winds due to absorption of solar radiation by water vapour in the troposphere and ozone in the stratosphere and also due to latent heat release in the troposphere. There are also tides produced in situ in the thermosphere due to extreme ultraviolet light absorption. The tides that move westward with apparent motion of the Sun are called the migrating tides. The migrating diurnal tidal characteristics in the stratosphere had been retrieved using temperature retrievals from Challenging Minisatellite Payload (CHAMP) observations during May 2001 to August 2005 (Zeng et al., 2008) and FORMOSAT-3/COSMIC mission using monthly data for the period 2007-2008 (Pirscher et al., 2010). Maximum amplitudes of 0.8 to 1.0 K were found over the tropics at 30 km altitude  in both studies. There are several papers in literature that describe the theory (Chapman and Lindzen, 1970; Forbes and Garrett, 1979) and observed characteristics of tides at various altitudes in the stratosphere, mesosphere and thermosphere from ground based measurements of radars and lidars (Liu et al., 2007; Pancheva and Mukhtarov, 2000; She et al., 2004; Xue et al., 2007; Baumgarten and Stober, 2019), satellite observations of TIMED Doppler Interferometer (TIDI) and Sounding of the Atmosphere using Broadband Emission Radiometry (SABER) instruments onboard TIMED (Mukhtarov et al., 2009; Wu et al., 2006), from Upper Atmosphere Research Satellite (UARS) (Shepherd et al., 2012; Wu et al., 1998), MLS (Wu and Jiang, 2005), reanalysis (Gan et al., 2014) and model datasets (Sakazaki et al., 2018; McCormack et al 2017). Based on results obtained from TIMED tidal diagnostics, the Climatological Tidal Model of the Thermosphere (CTMT) constituting the most important diurnal and semidiurnal tides have been proposed (Oberheide et al., 2011a). Using global cloud imagery, the Global Scale Wave Model (GSWM) was developed for tides arising due to latent heat releases (Hagan, 2002; 2003). Such models are further used as parameterizations for other global circulation models in the lower and upper atmosphere.

There are also the non-migrating tides in the atmosphere whose apparent motion is either slow or faster than the Sun. Some of these tides are thought to be produced due to non–linear interactions between stationary planetary waves (SPW) and migrating tides. However, significant debate exists if the non–migrating tides are truly a geophysical phenomenon or are an artefact of the method of analysis. It was proposed that SPW of wavenumber 1 (SPW1) interacts non–linearly with diurnal

migrating tide (DW1) and results in the non-migrating tides DS0 and DW2 [(The notation of the tides is as follows: First letter indicates the period of the tide - D for diurnal, S for semi-diurnal, T for terdiurnal; second letter indicates if the tide is westward (W) or eastward (E) propagating or stationary (S), and finally the last character is a digit which gives the wavenumber of the tide. The same notation will be followed for the rest of the paper]). Similarly, (SPW1) interacts with semi-diurnal migrating tide (SW2) and produces SW1 and SW3. Many studies support this school of thought based on correlation studies (Xu et al., 2014). However, it is also a possibility that  a high correlation is observed because of aliasing between these different components.

Among the non-migrating tides, a reasonably well understood tide is DE3 (eastward propagating diurnal tide of wavenumber 3). The observation of the wave-4 structure in the equatorial ionisation anomaly of the ionosphere due to DE3 tide is one of the most important discoveries of the last decade (Immel et al., 2006). The DE3 tide is very unique to the Earth and is produced in the troposphere due to the specific distribution of the landmasses and oceans and associated heating (Oberheide et al., 2011b). As the tide propagates upwards it modifies the various atmospheric parameters and this emphasizes the importance of troposphere-ionosphere coupling and also the need for obtaining the short term tidal variabilities.

The various tides generated in the lower atmosphere propagate upward, grow in amplitude and affect the large scale dynamics, chemistry and energetics of the thermosphere and ionosphere. Thus accurate determination of the variability of these various tides and other waves at the point of generation is extremely important to understand the atmospher coupling processes. In the current study, temperature data from FORMOSAT-3/COSMIC during 2009 to 2010 is analysed to extract migrating and non-migrating tides and stationary planetary waves globally over shorter time periods of ±10 days. Along with  diagnosing the short term variability in the said tides, the paper also addresses the aliasing involved between (1) mean temperature and migrating tides and (2) stationary planetary waves and non-migrating tides, particularly in the high latitudes. The paper is organised as follows. Section 2 describes the FORMOST-3/COSMIC data used, satellite sampling and phase space of the various wave components. The data analysis method of least square fitting is described briefly in Section 3. Tidal characteristics and associated aliasing are described in Sections 4 & 5, respectively, and the results are discussed and summarised in Section 6.

**2 Data and Sampling**

COSMIC is a constellation of six micro satellites working on the principle of Global Positioning System Radio Occultation (GPS RO) (Anthes et al., 2008). It involves active Earth limb sounding by radio transmissions by GPS satellites at 20,200 km and are observed by the COSMIC satellites in low Earth orbits (Anthes et al., 2008). The phase delay of L1 and L2  signals received is due to change in refractivity which is converted to electron density in the ionosphere and temperature and other parameters in the lower atmosphere and are described in detail in literature (Kuo et al., 2004; Kursinski et al., 1997). Briefly, the Earth's refractive index at microwave wavelengths is affected by the dry neutral atmosphere, water vapour and free electrons in the ionosphere and thus by deriving the refractivity of the atmosphere, the above mentioned parameters can be

retrieved. This technique provides a near-vertical scan of the atmosphere with good vertical resolution, global coverage, and insensitivity to atmospheric particulate matter (Kuo et al., 2004; Kursinski et al., 1997). The six satellites are have been placed in ~800 km orbits with 30° separations. This enables the local time coverage of all satellites, taken together, theoretically, over any given location to be possible in approximately 10 days. In this way, COSMIC satellites have a huge advantage over SABER in terms of global coverage. However, the altitude coverage of COSMIC is from surface to 60 km ('atmPrf', - dry temperature data product) with temperatures data reliable up to 50 km over the equator and further lower over mid and high latitudes (Das and Pan, 2014)., while In Contrast, SABER has coverage from 20 to 120 km, and thereby the latter which enables studies of stratosphere, mesosphere and lower thermosphere. Thus the data from COSMIC can only be used for tropospheric and stratospheric studies (and the ionospheric data products can be used for ionospheric studies). In the current study, level 2 dry temperature 'atmPrf' profiles from the lower atmospheric data from FORMOSAT-3/COSMIC mission are analysed for the period from October 2009 to December 2010. Data is considered at 1 km intervals from 15 to 50 km. It is known that the vertical resolution of RO derived temperature profiles is 0.5 km in the troposphere and 2 km in the stratosphere (Kursinski et al., 1997; Scherllin-Pirscher et al., 2017). COSMIC temperatures are smaller by 2 to 3 K than SABER temperatures across all latitudes below 0.3 hPa and larger above this altitude. The agreement of COSMIC temperatures with those from Aura MLSMicrowave Limb Sounding instrument onboard the Aura satellite observations is much better and in the range of ±1 K upto 2 hPa (Das and Pan, 2014).

**2.1 Satellite Sampling and Phase Space:**

The data obtained from COSMIC using the technique of GPS RO are not regular, i.e., the retrieved data are not uniformly spaced in space and time. This non-uniform and pattern-less spatial and temporal sampling is in a way supportiveadvantageous to the current study to characterise the variability of tides in the middle atmosphere as the method of least squares fitting would beis used. SABER onboard TIMED satellite is in a near Sun synchronous orbit and its observations are regular and in a clustered mode. Over a given location, the satellite observes the atmosphere at gradually reducing local times, and takes sixty days to cover the entire 24 hour diurnal duration. Due to this, to obtain sampling at all local times using SABER, analysis has to be done over data accumulated for 60 days. Thus, the tidal parameters obtained are a temporal average representation over the sixty days considered. On the other hand, accumulating data for 60 days induces serious aliasing problems as the variation in average or mean temperature aliases into the diurnal tides' amplitudes and the energy of SPWs alias into that of the non migrating tides. These problems are addressed here using COSMIC data.

As mentioned earlier 10 days data from all six COSMIC satellites is required in principle sufficient to cover appropriately sample the 24 local hour diurnal duration over any given location and thusallowing short term tidal variability can be obtainedto be diagnosed. If data from only one satellite is considered, one would require sixty days of data for tidal analysis, similar to SABER.

To establish this aspect and to ascertain the necessary and sufficient conditions for the amount of data required for accurate tidal characteristic extraction, COSMIC data is considered as follows for the analysis. Data is divided into two overlapping

130 groups, consisting of four satellites each. First group, named group 'G1', takes data from satellites C001, C002, C003, and C004 and the second group, named group 'G2', takes data from C004, C005, C006 and C001. (Data availability is shown in Figure S1, in supplementary section, as number of profiles available over equator, 30°N, 45°N and 65°N during the study period, from each of the COSMIC satellites C001 to C006. The last panel shows data available from all satellites taken together.) In principle, we could have divided the satellites into groups of three satellites and considered data over ~20 days, however,

135 due to technical problems, sometimes data from one or another of the satellites is not available entirely, or less data is available. To overcome this, we made groups of four, with two satellites in common and considered data over ±10 days centred over a given day. A third group consisting of all six satellites is also investigated; this is named group 'G0'. Further, data of G0 is  analysed by considering ±10 days' data centred over each day, to maintain uniformity and avoid data gaps. Differences observed in results obtained from G0, G1 and G2 allow the effect of aliasing to be examined and their role in

140 causing errors in diagnosed results evaluated. Data from the C004 satellite is also analysed separately using the same method by considering data over ± 30 days.

When satellite data are considered for tidal analysis, for minimal aliasing related problems, it is  important that the two dimensional space of universal time and longitude (over each latitude) is uniformly

145 sampled by the satellite. The same can be verified from a different perspective of total phase.  Given the universal time ($t$) and longitude ($\lambda$) of each observation,  the total phase is  for each wave of frequency ($f$) and wave number ($s$)  can range between 0 to $2\pi$. If all phases of a given wave are sampled, i.e., if phase sampling is sufficiently uniform, then the characteristics of the wave, namely, amplitude, and phase, can be extracted reasonably accurately. To understand this, the total phase of the important wave component DW1 is

150 investigated over the equator and 65°N and shown in Figure 1 for the different groups G0, G1 and G2 (by considering ±10 days' data) as well as for the C004 satellite (by considering ±30 days' data). It can be seen that for both latitudes , the phase sampling is reasonably uniform on any given day for all the waves. The number of data points are reduced in general over the period investigated from October 2009 to December 2010, due to reduction in overall number of observations. It can also be seen that the sampling is also uniform when data from one satellite (C004) was considered over ±30 days.  The

155 completeness and uniformity of the phase space sampling was also verified for all other waves of interest to the current study.

**3 Analysis**

Data in each group (G0/G1/G2/C004) are investigated using the least squares fitting technique. The following function is fit to the two dimensional temperature data, $T$, at each altitude at universal time, $t$, and longitude, $\lambda$,  to include (a) mean

160 temperature variation ($T_0$), (b) diurnal (frequency, $f_1 = 1$), semi-diurnal ($f_2 = 2$) and ter-diurnal ($f_3 = 3$)) tides with wave numbers $s_j$ ranging from -4 to 4, where negative wave numbers denote eastward propagating tides and positive wave numbers denote westward propagating tides and (c) SPWs with wave numbers $s_k$ ranging from 1 to 3.

$$T(t,\lambda) = T_0 + \sum_{i=1}^{3}\sum_{j=-4}^{4} T_{ij}\cos(2\pi f_i t + 2\pi s_j \lambda - \phi_{ij}) + \sum_{k=1}^{3} T_k\cos(2\pi s_k \lambda - \phi_k)$$

wWhere, $T_{ij}$ and $\phi_{ij}$ are the amplitudes and phases of the tides, and $T_k$ and $\phi_k$ are the amplitudes and phases of the SPWs.

165 It may be noted that data from G0, G1, G2 are analysed using ±10 days' data and data from C004 is analysed using ±30 days' data. This equation results in 61 fitted parameters that are carefully investigated in the ensuing sections.

**4 Tidal Characteristics**

The mean temperature and amplitudes of DW1 and SPW1 at 30 km obtained from the analysis of temperature data during November 2009 – September 2010 are shown in the three rows of Figure 2, respectively. Each column indicates the results
170 obtained from the three groups G0, G1, and G2 using ±10 days data and from satellite C004 using ±30 days data. The last column shows the numerical difference between results obtained from group G0 and C004. It can be seen that the results obtained from the three groups are very similar, with minute extremely small differences over very fine scales. The variation in the mean temperature is similar in all groups. Over the equator a strong semi-annual variation is observed (with maximum maxima during November and May) along with an annual variation with a maximum during April-May and a minimum during
175 November-December. and oOver mid and high latitudes a strong annual variation is observed with a maximum during summer and a minimum during winter. The sudden stratospheric warming (SSW) of 2010 is also observed in the northern hemisphere during January - February.  The migrating diurnal tide, DW1, is very prominent at 30 km over the equatorial region with amplitudes in the range 1-1.5K and showing as a band like structure around the equator. The band is slightly shifted towards winter poles, i.e., northward during northern hemisphere winter and southward during southern hemisphere winter. It may be
180 noted Note that amplitudes below 0.5 K are not shown in the figure. Over At latitudes greater than 45° in the winter hemisphere, intermittent patches of DW1 are observed.

The amplitudes during January 2010 are particularly larger enhanced relative to other times and in the range of 2-3 K. This coincides which coincide with the occurrence of the SSW of 2010. The SPW1 shows very large amplitudes are also large, reaching 18 K, over mid-latitudes beyond above 45° in the winter hemisphere. Specifically, the The amplitude of this wave is
185 stronger in the northern hemisphere than in the southern hemisphere. Furthermore, there is an apparent 60-day modulation in the amplitude of this wave by, what seems to be, a planetary wave of period ~60 days. These plots show that similar results are obtained with all groups. that all groups give similar results and thereby, Hence for the rest of the paper only analysis from group G0 (that considers data from all six satellites) is only discussed.

Results obtained using data over ±30 days from one the single satellite, C004, are very different from the group results,
190 particularly in the mid and high latitudes. The mean temperature is smoothened smoother over the 60 day period and the

difference between mean temperature of group G0 and C004 (presented in the last column of Figure 2) shows periodic variations of ~60 days over the entire global region. The differences maximise during winter and spring in the high latitudes with magnitudes greater than 3K. The amplitude of DW1 over the equator and  latitudes  less than 30° are similar to those obtained from the analysis of the groups, However, the values are unusually large over mid and high latitudes, particularly over the regions poleward of 45°. The differences show that DW1 amplitudes from C004 using ±30 days data are overestimated by more than 6K, which is significant, given that the maximum amplitudes of DW1 (from the group analysis) in high latitudes is less than 3 K. The amplitude of SPW1 is, on the other hand, similar to the variation observed in the analysis of the data in groups, However, the former is smoothe over the time duration considered. Here , the difference panel in the last column shows that the SPW1 amplitude observed by data from C004 alone is also modulated by periodic variations of ~60 days, particularly in the high latitude winter atmosphere. These differences are  of the order of ±3 K, which are small compared to the maximum SPW amplitudes.

Figure 3 shows the annual mean of the various wave parameters of interest in the current study using group G0 and satellite C004. The annual mean of mean temperature is similar in both columns. However, the migrating diurnal tide as well as nonmigrating tides are overestimated by C004 in the high latitudes. Over equator and low latitudes, the tidal amplitudes are similar.  SPW1 amplitudes are marginally underestimated by C004 over high latitudes. This could be due to the effect of smootheing as more data was used in the analysis of the latter.

Figure 4 shows the variation of amplitudes and phases of DS0 (left column), DW1 (middle column) and DW2 (right column) during the winter of 2009/10, i.e., from December 2009 to February 2010, over 65°N in the first and second rows and over equator in the third and fourth rows, respectively. These results are obtained from group G0.  In the high latitudes  winter hemisphere, DW1 shows large amplitudes of 2K, but only intermittently . DS0 and DW2 also show similar intermittent behaviour  in the range 1-2 K. The phase plots of these tides do not show any specific pattern as the waves themselves are intermittent. Over the equator, the amplitude of DW1 maximises at 30 km and is in the range of 1-1.5 K. Its phase variation with altitude indicates that its wavelength is ~25 km as is known from previous studies. Small amplitudes of 0.5 to 1 K are observed for DS0 and DW2 on either side of this equatorial band at 35 km, However, at all other altitudes over the equator and low latitudes their amplitudes are zero (not shown here). ~~The DW1 maximum at 30 km over equator peaks at noon indicating that the observed tidal variation is due to ozone absorption and is in phase with the sun. The altitude variation shows that the vertical wavelength of DW1 is ~25 km as known from previous studies.~~

Figure 5 shows the variation of the amplitude and phase of SPW1 at various altitudes from 20 to 50 km (along the different rows) during the period of study. Large amplitudes of SPW1 are seen in the high latitude winter atmosphere. The amplitudes of SPW1 over northern hemisphere are largest during winter, with values reaching 18 K at 30 km altitude. It is also seen that in January 2010, as the SSW started, the amplitude of SPW1 decreased to ~6K. In the southern hemisphere the amplitudes are smaller with a maximum of ~10 K at 30 km. Importantly, there is significant variability at a periodicity of ~60 days at all altitudes, the modulation remaining coherent between both hemispheres

225 Investigation of the phase plots show that the phase lines are nearly constant as a function of latitude at each height when large amplitudes of the SPWs are observed. The phase variation with altitude indicates that the vertical wavelength of SPW1 is in the range of 50 to 60 km at ~60°N, as well as at ~60°S.

Comparison of Figures 4 and 5 shows that significant amplitudes of DS0 and DW2 occur at the same time as when SPW1 is strong. However, there does not seem to be any significant correlation between the non-migrating tides and

230 the stationary planetary wave. To investigate this aspect further, Figure 6 shows the mean temperature in black, amplitudes of the DS0 in blue, DW1 in red, DW2 in cyan, and SPW1 in green at 30 km at 65°N, equator, and 65°S, in the three panels. The amplitudes of the waves are indicated by the axis on the right of the figure with that of SPW1 scaled down by a factor of 10 for convenience. The most striking feature of the figure is the occurrence of the SSW in January 2010 at 65°N. Exactly during the time when the mean temperatures were increasing at a high rate, the amplitude of

235 SPW reduced drastically from 17 to 10 K and the amplitudes of the three tides increased. The amplitude of DS0 is almost 2 K, that of DW1 is 1.5 K and that of DW2 is 1 K. Similar peaks are also observed after the event when the mean temperature is decreasing. At this point, the amplitude of DW1 maximises at 2 K, that of DS0 is 1 K and the amplitude of DW2 is less than 1K. During summer there is no wave activity in either hemispheres. In the southern hemisphere at 65°S, tidal activity is observed as the temperatures

240 start to rise in the winter. Particularly, as the mean temperature increases rapidly during July 2010, amplitude of DW1 maximises at ~2K and the amplitudes of DS0 and DW2 are of the order of 1 K. During the next 3-4 months, intermittent patches are observed when the amplitudes of DW1 and DW2 are ~1.5 K. Over the equator, on the other hand, the picture is very simple as there are no occurrences of significant amplitudes of DS0 and DW2. Interestingly, DW1 shows significant short term variability at periodicities of the order of 30

245 days. If the data were analysed over 60 day intervals, this variability would not have been observed. The amplitudes of DW1 are marginally higher (~ 1.5 K) during northern hemisphere winter and smaller (~1K) during summer.

To understand the simultaneous occurrence of non-migrating tides and stationary planetary waves, a simple correlation study is performed and is shown in Figure 7. The first panel shows the correlation of DS0 and SPW1 at 65°N at 30 km for winter from December 2009 to February 2010. The correlation is negligible. The

250 second panel shows the correlation between DW2 and SPW1 for the same latitude and altitude. Here also the correlation is not significant. The bottom panels show similar correlations for the southern hemisphere during June to August 2010 and there is good correlation between DS0 and SPW1 and a reasonable correlation between DW2 and SPW1. However, the amplitudes of the tides are all ~1 K or smaller. Thus no reasonable statistical relation can be established between occurrence of non-migrating tides and stationary planetary waves.

255 Figure 8 shows another correlation study between mean temperature and DW1 tide. In the left panels no significant correlation between the two parameters in either hemisphere is present. The panels on the right show the variation of DW1 as a function of gradient in the mean temperature. When the latter are larger than ±0.25 K day$^{-1}$, the amplitudes of DW1 are also very large and increasing with increasing

gradient. The situation is same in the southern hemisphere too. This effect is not observed when the gradients are smaller and then at these times the amplitudes of DW1 are smaller than 0.5 K, and are negligible. This clearly indicates that there is an aliasing of energy into the DW1 tidal amplitude when the mean temperatures vary significantly.an aliasing of energy due to varying mean temperature into the DW1 tide amplitude.

**5 Aliasing**

It was very clearly established that varying mean temperatures alias into the DW1 tide using SABER data (Sakazaki et al., 2012). However, in the case of COSMIC data, as the data sampling is irregular, it is difficult to establish such aliasing phenomena in the same way. To circumvent this problem, numerical experiments weare performed to understand the extent to which aliasing occurs as a result of amount of aliasing occurring with respect to COSMIC data sampling. For the times and locations of COSMIC measurements over the equator and at 65°N, a fake numerical atmosphere is created that consists of known variabilities. Table 1 describes the 10 such cases considered for this study. The results from these numerical experiments are shown in Figures 9 and 10 and are explained in detail in the table.

Table 1: Numerical Experiments to investigate the aliasing of energy from one component into another over equator and 65°N as seen in Figures 9 and 10. T is temperature, $\lambda$ is longitude, t is day number and time, and h is hour of day.

| S. No | Atmosphere | Equation | RemarksResults |
|-------|-----------|----------|---------|
| 1 | Constant SPW1 Amplitude | $T = 10 * \cos(2\pi\lambda/360)$ | SPW1 amplitude extracted with no errors over both latitudes |
| 2 | Slowly varying SPW1 amplitude (Period = 50 days) | $T = 10 + [10 * \cos(2\pi t/50)] * \cos(2\pi\lambda/360)$ | Maximum amplitude is underestimated and minimum amplitude is overestimated. No Aliasing is observed over equator, however, over 65°N, DS0 and DW2 components show equal and uniform aliasing of 1 K amplitudes. This happens at times of maximum gradient in SPW1 amplitude. |
| 3 | Fast Varying SPW1 amplitude (Period = 10 days) | $T = 10 + [10 * \cos(2\pi t/10)] * \cos(2\pi\lambda/360)$ | Average SPW1 amplitude is extracted. The periodic variation of 10 days is lost in the analysis. However, no DS0 and DW2 components are observed over both latitudes. |

| 4 | Constant DS0 Amplitude | $T = 10 * \cos(2\pi h/24.)$ | Constant DS0 component extracted successfully. No aliasing into any other component is observed over both latitudes. |
|---|---|---|---|
| 5 | Varying and large DS0 Amplitude (Period = 10 days) | $T = 10 + [10 * \cos(2\pi t/10)] * \cos(2\pi h/24.)$ | Average DS0 component is extracted. And no aliasing into other components is seen over both latitudes. |
| 6 | Varying and Small DS0 Amplitude (Period = 5 days) | $T = 1 + [1 * \cos(2\pi t/5)] * \cos(2\pi h/24.)$ | Average DS0 component is extracted. And no aliasing into other components is seen over both latitudes. |
| 7 | Varying and Small DW2 Amplitude (Period = 5 days) | $T = 1 + [1 * \cos(2\pi t/5)] * \cos(\frac{2\pi h}{24}. + \frac{2\pi \lambda}{360} 2)$ | Average DW2 component is extracted. And no aliasing into other components is seen over both latitudes. |
| 8 | Fast Varying Mean Temperature (Period = 10 days) | $T = 280 + 10 * \cos(2\pi t/10.)$ | Average mean variation is extracted. Aliasing is observed in DW1, SPW1 and DS0 and DW2, all showing amplitudes up to 1 K over equator and larger  at 65°N. The large peaks observed in DW1 and SPW1  at 65°N indicate significant aliasing. |
| 9 | Slowly Varying Mean Temperature (Period = 50 days) | $T = 280 + 10 * \cos(2\pi t/50.)$

 (Similar to variation during SSW 2010) | Mean variation is extracted reasonably (in the range 273 to 287 K), with maximum amplitudes underestimated by 3 K and minimum values overestimated by 3 K. Aliasing is observed in DW1, SPW1 and DS0 and DW2, all showing amplitudes up to 1 K over equator and much larger  at 65°N. Large peaks of 5 K are observed in DW1 and that of 3 K are observed in SPW1  at 65°N. Interestingly, the peaks in DW1 occur at times of maximum gradient in mean temperature |
| 10 | Constant Mean Temperature | $T = 280$ | Mean temperature is extracted and absolutely no aliasing is observed over both latitudes. |

275

**6 Discussion and Concluding Remarks**

Extraction of tidal variability from satellite measurements with good accuracy and with no aliasing is a challenge. Using SABER temperature data, wave characteristics can be extracted over 60 days in the middle atmosphere. The amplitude of SPWs using SABER data are much smaller (Xu et al., 2014) than those obtained in the current study. During northern hemisphere winter, the maximum average amplitudes from SABER were 7.2±1.02 K at 45°N and 45 km. There was strong temporal correlation between the occurrence of SPWs and the non-migrating tides, which led to the conclusion that the latter were produced due to non-linear interactions of SPWs and migrating diurnal tides (Xu et al., 2014). Their study concentrated explicitly upon the generation of these nonmigrating tides and hence their conclusions. However, the current study shows that the amplitude of SPW1 is very large, of the order of 18K, and the strong temporal correlation with DS0 and DW2 could also be caused due to aliasing of SPW1 into the non-migrating tides. In Case 2 of the numerical experiments, it is observed that the aliasing of SPW1 into DS0 and DW2 is equal and uniform and thus in the actual analysis if DS0 and DW2 are found to be equal and uniform, it is possible that the diagnosed variation in these tidal components might be due to aliasing. Thus the question of whether non-linear interactions between SPW1 and DW1 producing DS0 and DW2 is still debatable. Although non-linear interactions cannot be entirely ruled out, the current study shows that the contribution of this mechanism in producing nonmigrating tides in the mid and high latitude stratosphere is not as important as indicated by earlier studies, that are particularly dependent on analysis of SABER data (Xu et al., 2014). The current study indicates that the DS0 and DW2 components are much smaller than those observed earlier using SABER data.

Baumgarten and Stober (2019) derived short term tidal variability in the altitude range from 30 to 70 km using temperature derived from lidar observations at Kühlungsborn (54°N, 12°E), a mid latitude station. The diurnal tide (including all wavenumbers) in temperature and winds was extracted from lidar data and compared with the DW1 component of temperature and winds from Modern-Era Retrospective analysis for Research and Applications, Version 2 (MERRA-2). It was shown that the local tidal fields are dominated by the migrating diurnal and migrating semidiurnal tides and that other components are negligible. This indicates that the nonmigrating components make a small contribution to net tidal fields and thus supports the conclusions of the current study that diagnosed non-migrating tidal signatures could be possibly due to aliasing.

Aliasing problems involved in SABER data are difficult to verify due to lack of similar global observations but comparisons have been made with models and reanalysis and it was noted that there are significant inconsistencies in the tidal signatures determined from the various sources (Sakazaki et al., 2018). It was found that the amplitude of the trapped diurnal migrating tide in the upper stratosphere is significantly smaller in reanalyses than that in SABER. The current study also indicates that SABER tidal amplitudes are overestimated, particularly in the mid and high latitudes. Results from space time spectral analysis of gridded monthly COSMIC data for the period from 2007 to 2008 also had shown that the DW1 peaks at 30 km over the equatorial latitudes (Pirscher et al., 2010). It was shown

in their paper that sampling was insufficient northward of 50° and the spectral amplitude  associated with the sampling
310 error was large. However, in the current study (where a different time interval and hence different distribution of satellite observations was selected) the wave phase space  is sufficiently well sampled (Figure 1). By using the least squares method over shorter lengths of data, it was possible to extract the different wave components. The numerical experiments show that with the given sampling and the technique used, it can be verified whether the extracted spectral components SPW1, DS0, DW1, DW2 are geophysical or are a result of aliasing.

315 There are also studies that have shown that the time evolution of DW2 over the equatorial mesopause region follows SPW1 variations in the high latitude stratosphere (Lieberman et al., 2015; Niu et al., 2018). It is proposed that mid to high latitude stratospheric SPWs are ducted upward and equatorward, interact with equatorial DW1 over mesopause and thereby generate DW2 over the equatorial mesoapuse region. DS0 is not  discussed by Lieberman et al (2015). Niu et al (2018) investigated the SPW1-DW1 interaction during SSWs using the extended Canadian Middle Atmosphere Model (eCMAM) data and found
320 good but varying degrees of correlations with DS0 and DW2 during 20 out of 31 SSW events indicating that the strength of non-linear interactions also varied from year to year. As the correlations are not observed during all SSW events the proposed mechanism of non-linear interactions is still unproven.

In the current study, during the SSW of 2010, the peaks observed in DW1 and DS0 & DW2 are most likely due to aliasing. At 65°N, as the temperature increases (decreases) steeply during the onset (decay) of the warming
325 episode, the DW1 component is observed to be large (1.5 to 2 K). The entire SSW event lasted  ~60 days  and the temporal evolution of its fields is very similar to the numerical experiment in Case 9, where significant aliasing into DW1 is observed. This experiment indicates that over high latitudes, when there is a large gradient in the mean temperature, peaks of large amplitude of DW1 are observed, which are not geophysical in nature. At the same time, the SPW1 component steeply  decreased during the onset of the episode, from which the DS0 and DW2 components
330 may have arisen due to aliasing. In addition, there is  aliasing  to SPW1 of the order of 2-3 K, but since the observed SPW1 amplitudes are much larger (18 K), this is of less geophysical significance. The amplitude of 2K of DS0 during the onset of the event may have some geophysical significance, but  further investigation is needed before this is clear.

McCormack et al (2017) investigated the short term tidal variability during the SSWs of January 2010 and January 2013 using
335 high altitude Navy Global Environmental Model (NAVGEM) data in the mesosphere and lower thermosphere region. NAVGEM is the result of assimilation of middle atmospheric data from nine meteor radar stations and other satellite instruments, including SABER on board TIMED satellite. Their results show a reduction in the semi-diurnal amplitude before the onset of the SSW and an increase after the event, peaking 10-14 days later.

Short term tidal variability  has also been deduced  using data from a Sodium Lidar and simultaneous SABER retrievals
340 and TIME-GCM results in the mesosphere and lower thermosphere (Liu et al., 2007). They found large tidal variability which could be the result of interactions with the planetary waves. The migrating diurnal tidal amplitude was modulated by the planetary wave of 5-7 day period. Such interactions are worth studying in the future using COSMIC data by considering

travelling planetary waves to obtain more insights into the tidal variability. Unfortunately, the altitude coverage of COSMIC is only up to the stratopause and thus tidal characteristics cannot be extracted for altitudes above 45 to 50 km. However, the current study clearly establishes the fact that with COSMIC data short term tidal variability can be obtained in combination with consideration of the aliasing involved. The following may thus be concluded from the current study.

1. COSMIC data is better suited for tidal studies than along track observations from a single satellite due to better phase sampling of tides and waves; however, due to the lack of altitude coverage the studies are confined only to the lower stratosphere.

2. The migrating diurnal tide (DW1) is found to maximise at 30 km over the equator its seasonal variation in latitude is attributed to the excitation of more than one tidal mode in the troposphere.  The vertical wavelength is of the order of 25 km.

3. A stationary planetary wave of wave number one (SPW1) peaks in the winter hemisphere over high latitudes with a vertical wavelength of 50-60 km at 65°N. It exhibits a strong ~60 day variability which was not observed earlier in SABER studies.

4. DS0 and DW2 components are relatively small and only  present intermittently in the high latitude middle atmosphere COSMIC analysis. Most of the peaks seem to be appearing due to aliasing.

5. Aliasing is significantly reduced when data is analysed over ±10 days using COSMIC data. However, it still exists and the numerical experiments performed in the current study show that DS0 and DW2 components arise when there is a rapid change in the SPW1 amplitude over time. Similar aliasing into DW1 component is prominently observed when there is a rapid change in the mean temperature, particularly in the high latitudes.

6. These exercises indicate that at the time of the SSW in January 2010, the peaks observed in DW1, as well as DS0 and DW2, are likely a manifestation of the aliasing effects involved in satellite data analysis, and that they may not be geophysical.  Analyses of satellite data needs to be done extremely carefully  when identifying the various tidal components and their characteristics.

It is thus concluded that non-linear interactions are not a very important source of generation of nonmigrating tides in the winter high latitude stratosphere.

**Code Availability**

The codes are prepared in IDL and can be supplied on request.

**Data availability**

Data used in the current study is obtained from UCAR/COSMIC. The data is freely available.

**Author contribution**

375 UD and WW conceived the idea.

UD performed the data analysis.

CJP provided insights into usage of COSMIC data.

WW designed and SKD performed the numerical experiments.

UD and WW analysed and finalised the results after discussion with all authors.

380 UD prepared the manuscript with contribution from all authors.

**Competing interests**

The authors declare that they have no conflict of interest.

**Acknowledgements**

Authors acknowledge UCAR/COSMIC program for providing free access to FORMOSAT-3/COSMIC 'atmPrf' temperature
385 data. UD is supported by Early Career Research Award ECR/2017/002258 by Science and Engineering Board (SERB), Govt.
of India. Support for portions of this work came from National Science and Engineering Research Council (NSERC)
Collaborative Research and Training Experience Grant (384996-2010) for CANDAC activities. WEW was supported by an
NSERC Discovery Grant. CJP is supported by the Ministry of Science and Technology of Taiwan through the grant MOST-
107-2111-M-008-006.

390

**Figures**

[Figure]

395   Fig 1.   Distribution of Pphase space for the tide DW1 from groups G0, G1, G2 (±10 days' data) and for the satellite C004 (±30 days' data) during the study period (2009-2010) over the equator and 65°N.

[Figure]

Fig 2.   Rows show the variation in mean temperature, amplitudes of the DW1 tide and SPW1 wave, respectively. Columns

400   are results from groups G0, G1, and G2, and from satellite C004, respectively, for 30 km altitude. The last column is the numerical difference in results obtained from group G0 (±10 days' data) and C004 (±30 days' data).

[Figure]

Fig 3. Annual means of mean temperature ($T_o$) and amplitudes of diurnal tides (DS0, DW1, DW2) and stationary planetary wave (SPW1) for group G0 and satellite C004 during 2010 (Jan to Dec). Note the overestimation of amplitudes of the migrating tide, particularly in mid and high latitudes in the analysis of data over ±30 days using a single satellite (C004).

405

[Figure]

Fig 4.    Variation of amplitudes and phases of DS0 (left column), DW1 (middle column) and DW2 (right column) during the winter of 2009/10, i.e., from December 2009 to February 2010 over 65°N in the first and the second rows and equator in the third and fourth rowstop two and the bottom two rows, respectively.

[Figure]

Fig 5.    Variation of amplitude and phase of SPW1 at various altitudes from 20 to 50 km, as a function of latitude and day of the year.

[Figure]

Fig 6.    Mean temperature (black), amplitudes of the DS0 (blue), DW1 (red), DW2 (cyan), and SPW1 (green) at 30 km at

65°N, equator, and 65°S during the study period. The amplitude of SPW1 is scaled down by a factor of 10 for

convenience.

[Figure]

Fig 7.    Correlation between DS0 & SPW1 and DW2 & SPW1 during winters at 65° latitudes in the northern and southern hemispheres.

[Figure]

425    Fig 8.    Correlation between DW1 & mean temperature and DW1 & gradient in mean temperature during winters at 65° latitudes in the northern and southern hemispheres.

[Figure]

Fig 9.    Results of numerical experiments from cases 1 to 7 (Table 1) for atmospheres considered to have only one variability among SPW1, DS0 and DW2.

430

[Figure]

Fig 10. Results of numerical experiments from cases 8 to 10 (Table 1) for atmospheres considered to have only a mean temperature variation.

**References**

435 Anthes, R. A., et al.: The COSMIC/FORMOSAT-3 Mission: Early Results, Bull. Am. Meteorol. Soc., 89(3), 313-333, doi:10.1175/BAMS-89-3-313, 2018.

Baumgarten, K., Gerding, M., Baumgarten, G., and Lübken, F.-J.: Temporal variability of tidal and gravity waves during a record long 10-day continuous lidar sounding, Atmos. Chem. Phys., 18, 371–384, https://doi.org/10.5194/acp-18-371-2018, 2018.

440 Baumgarten, K. and Stober, G.: On the evaluation of the phase relation between temperature and wind tides based on ground-based measurements and reanalysis data in the middle atmosphere, Ann. Geophys., 37, 581–602. https://doi.org/10.5194/angeo-37-581-2019, 2019.

Chapman, S., and Lindzen, R. S.: Atmospheric Tides, in Thermal and Gravitational, edited, Springer, Dordrecht, 1970.

Das, U. and Pan, C. J.: Validation of FORMOSAT-3/COSMIC level 2 "atmPrf" global temperature data in the stratosphere,
445    Atmos. Meas. Tech., 7(3), 731-742, doi:10.5194/amt-7-731-2014, 2014.

Forbes, J. M. and Garrett, H. B.: Theoretical Studies of Atmospheric Tides, Reviews of Geophysics and Space Physics, 17(8),
    1951-1981, 1979.

Forbes, J. M., Kilpatrick, M., Fritts, D. Manson, A. H., and Vincent, R. A.: Zonal mean and tidal dynamics from space: an
    empirical examination of aliasing and sampling issues, Ann. Geophys., 15(9), 1158-1164, 1997.

450    Gan, Q., Du, J., Ward, W. E., Beagley, S. R. , Fomichev, V. I., and Zhang, S.: Climatology of the diurnal tides from eCMAM30
    (1979 to 2010) and its comparison with SABER, Earth Planets and Space, 66:103, doi:10.1186/1880-5981-66-103, 2014.

Hagan, M. E. and Forbes, J. M.: Migrating and nonmigrating diurnal tides in the middle and upper atmosphere excited by
    tropospheric latent heat release, J. Geophys. Res., 107, D24, 4754, doi:10.1029/2001JD001236,2002

Hagan, M. E. and Forbes, J. M.: Migrating and nonmigrating semidiurnal tides in the upper atmosphere excited by tropospheric
455    latent heat release, J. Geophys. Res., 108, A2, 1062, doi:10.1029/2002JA009466, 2003.

Immel, T. J., Sagawa, E., England, S. L., Henderson, S. B., Hagan, M. E., Mende, S. B., Frey, H. U., Swenson, C. M., and
    Paxton, L. J.: Control of equatorial ionospheric morphology by atmospheric tides, Geophys. Res. Lett., 33, L15108,
    doi:10.1029/2006GL026161, 2006.

Kuo, Y. H., Wee, T. K., Sokolovskiy, S. Rocken, C., Schreiner, W., Hunt, D., and Anthes, R. A.: Inversion and Error Estimation
460    of GPS Radio Occultation Data, J. of the Meteorological Society of Japan., 82, 1B, 507-531, 2004.

Kursinski, E. R., Hajj, G. A., Schofield, J. T., Linfield, R. P., and Hardy, K. R.: Observing Earth's atmosphere with radio
    occultation measurements using the Global Positioning System, J. Geophys. Res.102(D19), 23429-23465, 1997.

Lieberman, R.S.; Riggin, D.M.; Ortland, D.A.; Oberheide, J.; Siskind, D.E.: Global observations and modeling of
    nonmigrating diurnal tides generated by tide-planetary wave interactions, J. Geophys. Res. Atmos., 120, 11419–11437,
465    doi:10.1002/2015JD023739, 2015.

Liu, H. L., Li, T., She, C. Y., Oberheide, J., Wu, Q., Hagan, M. E., Xu, J., Roble, R. G., Mlynczak, M. G., and Russell, J. M.:
    Comparative study of short-term diurnal tidal variability, J. Geophys. Res., 112, D18, doi:10.1029/2007JD008542, 2007.

McCormack, J., Hoppel, K., Kuhl, D., de Wit, R., Stober, G., Espy, P., Baker, N., Brown, P., Fritts, D., Jacobi, C., Janches,
    D., Mitchell, N., Ruston, B., Swadley, S., Viner, K., Whitcomb, T. and Hibbins, R.: Comparison of mesospheric winds
470    from a high-altitude meteorological analysis system and meteor radar observations during the boreal winters of 2009-2010
    and 2012-2013, J. Atmos. Solar-Terr. Phys., 154, 132-166, 2017.

Mertens, C. J., et al.: SABER observations of mesospheric temperatures and comparisons with falling sphere measurements
    taken during the 2002 summer MaCWAVE campaign, Geophys. Res. Lett., 31,
    L03105,doi:10.1029/2003GL018605,2004.

Mukhtarov, P., Pancheva, D., and Andonov, B.: Global structure and seasonal and interannual variability of the migrating diurnal tide seen in the SABER/TIMED temperatures between 20 and 120 km, J. Geophys. Res., 114, A02309, doi:10.1029/2008JA013759,2009.

Niu, X., Du, J., Zhu, X.: Statistics on nonmigrating diurnal tides generated by tide-planetary wave interaction and their relationship to Sudden Stratospheric Warming, Atmosphere, 9, 416, 2018.

Oberheide, J., Forbes, J. M., Zhang, X., and Bruinsma, S. L.: Climatology of upward propagating diurnal and semidiurnal tides in the thermosphere, J. Geophys. Res., 116, A11306, doi:10.1029/2011JA016784, 2011a.

Oberheide, J., Forbes, J. M., Zhang, X., and Bruinsma, S. L.: Wave-driven variability in the ionosphere-thermosphere-mesosphere system from TIMED observations: What contributes to the "wave 4"?, J. Geophys. Res., 116 A01306, doi:10.1029/2010JA015911, 2011b.

Oberheide, J., Hagan, M. E., Roble, R. G., and Offermann, D.: Sources of nonmigrating tides in the tropical middle atmosphere, J. Geophys. Res., 107(D21), 4567, doi:10.1029/2002JD002220, 2002.

Pancheva, D. and Mukhtarov, P.: Wavelet analysis on transient behaviour of tidal amplitude fluctuations observed by meteor radar in the lower thermosphere above Bulgaria, Ann. Geophys., 18, 316-331, 2000.

Pedatella, N. M., Oberheide, J., Sutton, E. K., Liu, H.-L., Anderson, J. L., and Raeder, K.: Short term nonmigrating tide variability in the mesosphere, thermosphere, and ionosphere, J. Geophys. Res. Space Physics, 121, 3621– 3633, doi:10.1002/2016JA022528, 2016.

Pirscher, B., Foelsche, U., Borsche, M., Kirchengast, G., and Kuo, Y. H.: Analysis of migrating diurnal tides detected in FORMOSAT-3/COSMIC temperature data, J. Geophys. Res., 115, D14108, doi:10.1029/2009JD013008, 2010.

Remsberg, E. E., Lingenfelser, G., Harvey, V. L., Grose, W., Russell, J. III., Mlynczak, M., Gordley, L., and Marshall, B.T.: On the verification of the quality of SABER temperature, geopotential height, and wind fields by comparison with Met Office assimilated analyses, J. Geophys. Res., 108, D20, 4628, doi:10.1029/2003JD003720, 2003.

Remsberg, E. E., et al.: Assessment of the quality of the Version 1.07 temperature-versus-pressure profiles of the middle atmosphere from TIMED/SABER, J. Geophys. Res., 113, D17101, doi:10.1029/2008JD010013, 2008.

Sakazaki, T., Fujiwara, M., and Shiotani M.: Representation of solar tides in the stratosphere and lower mesosphere in state-of-the-art reanalyses and in satellite observations, Atmos. Chem. Phys., 18, 1437-1456, https://doi.org/10.5194/acp-18-1437-2018, 2018.

Sakazaki, T., Fujiwara, M., Zhang, X., Hagan, M. E., and Forbes, J. M.: Diurnal tides from the troposphere to the lower mesosphere as deduced from TIMED/SABER satellite data and six global reanalysis data sets, Journal of Geophysical Research: Atmospheres, 117, D13108, doi:10.1029/2011JD017117, 2012.

She, C. Y., et al.: Tidal perturbations and variability in the mesopause region over Fort Collins, CO (41N, 105W): Continuous multi-day temperature and wind lidar observations, Geophys. Res. Lett., 31, L24111, doi:10.1029/2004GL021165, 2004.

Scherllin-Pirscher, B., Steiner, A. K.,Kirchengast, G.,Schwärz, M.,and Leroy,S. S., (2017), The power of vertical geolocation of atmospheric profiles from GNSS radio occultation, J. Geophys. Res. Atmos., 122, 1595–1616, doi:10.1002/2016JD025902.

510    Shepherd, G. G., et al.: The Wind Imaging Interferometer (WINDII) on the Upper Atmosphere Research Satellite: A 20 year perspective, Rev. Geophys., 50, RG2007, doi:10.1029/2012RG000390, 2012.

Wu, D. L., and Jiang, J. H.: Interannual and seasonal variations of diurnal tide, gravity wave, ozone, and water vapor as observed by MLS during 1991–1994, Adv. Space Res., 35, 1999-2004, doi:10.1016/j.asr.2004.12.018, 2005.

Wu, D. L., McLandress, C., Read, W. G., Waters, J. W., and Froidevaux, L.: Equatorial diurnal variations observed in UARS
515    Microwave Limb Sounder temperature during 1991–1994 and simulated by the Canadian Middle Atmosphere Model, J. Geophys. Res., 103, D8, 8909-8917, 1998.

Wu, Q., McEwen, D., Guo, W., Niciejewski, R., Roble, R. and Won., Y.-I.: Long-term thermospheric neutral wind observations over the northern polar cap, J. Atmos. Sol. Terr. Phys., 70, 2014-2030, doi:10.1016/j.jastp.2008.09.004, 2008.

520    Wu, Q., T. L. Killeen, D. A. Ortland, S. C. Solomon, R. D. Gablehouse, R. M. Johnson, W. R. Skinner, R. J. Niciejewski, and Franke, S. J.: TIMED Doppler interferometer (TIDI) observations of migrating diurnal and semidiurnal tides, J. Atmos. Sol. Terr. Phys., 68, 408-417, doi:10.1016/j.jastp.2005.02.031, 2006.

Xu, J., Smith, A. K., Liu, M., Liu, X., Gao, H., Jiang, G., and Yuan, W.: Evidence for nonmigrating tides produced by the interaction between tides and stationary planetary waves in the stratosphere and lower mesosphere, J. Geophys. Res., 119,
525    471–489, doi:10.1002/2013JD020150, 2014.

Xue, X., Wan, W., Xiong, J., and Dou, X.: Diurnal tides in mesosphere/low-thermosphere during 2002 at Wuhan (30.6°N, 114.4°E) using canonical correlation analysis, J. Geophys. Res., 112, D06104, doi:10.1029/2006JD007490, 2007.

Zeng, Z., Randel, W., Sokolovskiy, S., Deser, C., Kuo, Y.-H., Hagan, M., Du, J., and Ward, W.: Detection of migrating diurnal tide in the tropical upper troposphere and lower stratosphere using the Challenging Minisatellite Payload radio occultation
530    data, J. Geophys. Res., 113, D03102, doi:10.1029/2007JD008725, 2008.

Zhang, X., Forbes, J. M., Hagan, M. E., Russell, J. M., Palo, S. E., Mertens, C. J., and Mlynczak, M. G.: Monthly tidal temperatures 20–120 km from TIMED/SABER, J. Geophys. Res., 111, A10S08, doi:10.1029/2005JA011504, 2006.